# Mechanical basis and topological routes to cell elimination

**Siavash Monfared[1]\*, Guruswami Ravichandran[1], José Andrade[1], Amin Doostmohammadi[2]\***

[1]Division of Engineering and Applied Science, California Institute of Technology, Pasadena, United States; [2]Niels Bohr Institute, University of Copenhagen, Copenhagen, Denmark

**Abstract** Cell layers eliminate unwanted cells through the extrusion process, which underlines healthy versus flawed tissue behaviors. Although several biochemical pathways have been identified, the underlying mechanical basis including the forces involved in cellular extrusion remains largely unexplored. Utilizing a phase-field model of a three-dimensional cell layer, we study the interplay of cell extrusion with cell–cell and cell–substrate interactions in a flat monolayer. Independent tuning of cell–cell versus cell–substrate adhesion forces reveals that extrusion events can be distinctly linked to defects in nematic and hexatic orders associated with cellular arrangements. Specifically, we show that by increasing relative cell–cell adhesion forces the cell monolayer can switch between the collective tendency towards fivefold, hexatic, disclinations relative to half-integer, nematic, defects for extruding a cell. We unify our findings by accessing three-dimensional mechanical stress fields to show that an extrusion event acts as a mechanism to relieve localized stress concentration.

## Editor's evaluation

In this work, Monfared et al. construct a valuable three-dimensional phase-field model for cell monolayers and use this to investigate the relationship between single-cell extrusion events and topological defects in cellular arrangement. The extension of existing 2D phase field models to three dimensions is an important contribution of this paper, which will be of general interest to the theoretical modelling of epithelial monolayers. Here the model is used to study the importance of cell-cell and cell-substrate interaction in extrusion from cell monolayers, which will be of practical interest to biologists and physicists working on this process. This paper presents convincing evidence that extrusion events are distinctly linked to defects in nematic and hexatic orders in the cell monolayer.

\*For correspondence:
monfared@caltech.edu (SM);
doostmohammadi@nbi.ku.dk (AD)

**Competing interest:** The authors declare that no competing interests exist.

## Introduction

The ability of cells to self-organize and collectively migrate drives numerous physiological processes including morphogenesis (*Chiou and Collins, 2018*; *Vafa and Mahadevan, 2022*), epithelial–mesenchymal transition (*Barriga et al., 2018*), wound healing (*Brugués et al., 2014*), and tumor progression (*De Pascalis and Etienne-Manneville, 2017*). Advanced experimental techniques have linked this ability to mechanical interactions between cells (*Maskarinec et al., 2009*; *Ladoux, 2009*; *Ladoux and Mège, 2017*). Specifically, cells actively coordinate their movements through mechanosensitive adhesion complexes at the cell–substrate interface and cell–cell junctions. Moreover, cell–cell and cell–substrate adhesions seem to be coupled (*Balasubramaniam et al., 2021*), further complicating the interplay of mechanics with biochemistry.

While advances in experimental techniques are followed by more nuanced theoretical and computational developments, a majority of current approaches to simulate multicellular layers are limited

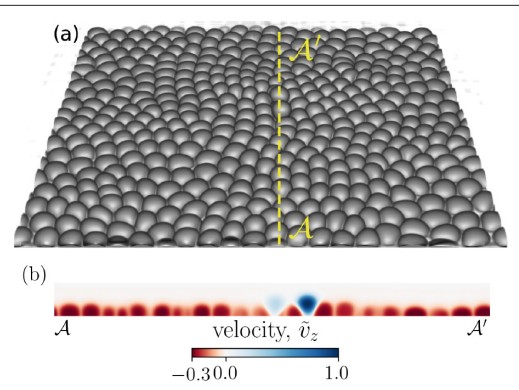

**Figure 1.** Cell extrusion in a 3D representation of a confluent cell layer. (**a**) A representative simulation snapshot (cell–substrate adhesion $\omega_{cw} = 0.0025$ and relative cell–cell adhesion $\Omega = \omega_{cc}/\omega_{cw} = 0.4$) of a three-dimensional cell monolayer. Two cells are visibly extruding. (**b**) A cross-section (dotted yellow line $\mathcal{A} - \mathcal{A}'$) of the cell monolayer highlighting the two extruding cells via the normalized out-of-plane velocity ($\tilde{v}_z = (\vec{v} \cdot \vec{e}_z)/v_z^{max}$), where $v_z^{max}$ is the maximum value of the $v_z$ component of the velocity field $\vec{v}$ in the shown cross-section.

to two-dimensional systems, hindering in-depth exploration of intrinsically three-dimensional nature of the distinct forces that govern cell–cell and cell–substrate interactions. Furthermore, some of the most fundamental processes in cell biology such as cell extrusion – responsible for tissue integrity – are inherently three-dimensional. Thus, studying the underlying mechanisms necessitates access to both in-plane and out-of-plane forces in the cell layers.

Cell extrusion refers to the process of removal of excess cells to prevent accumulation of unnecessary or pathological cells (**Rosenblatt et al., 2001**). This process can get initiated through apoptotic signaling (**Rosenblatt et al., 2001**), oncogenic transformation (**Hogan et al., 2009**), and overcrowding of cells (**Marinari et al., 2012**; **Eisenhoffer et al., 2012**; **Levayer et al., 2016**) or induced by replication stress (**Dwivedi et al., 2021**). Most importantly, cell extrusion plays an important role in developmental (**Toyama et al., 2008**), homeostatic (**Eisenhoffer et al., 2012**; **Le et al., 2021**), and pathological processes (**Slattum and Rosenblatt, 2014**), including cancer metastasis. However, the underlying mechanisms that facilitate cell extrusion are still unclear.

The similarities between cellular systems and liquid crystals, studied both theoretically and experimentally, featuring both nematic order (**Saw et al., 2017**; **Kawaguchi et al., 2017**; **Duclos et al., 2018**; **Blanch-Mercader et al., 2018**; **Tan et al., 2020**; **Zhang et al., 2021**) and hexatic order (**Classen et al., 2005**; **Sugimura and Ishihara, 2013**; **Pasupalak et al., 2020**; **Maitra et al., 2020**; **Hoffmann et al., 2022**) with the two phases potentially coexisting (**Armengol-Collado et al., 2022**) and interacting provide a fresh perspective for understanding cellular processes. The fivefold disclinations in hexatic arrangement of cells are numerically shown to favor overlaps between the cells in two-dimensions (**Loewe et al., 2020**), potentially contributing to the cell extrusion in three-dimensions. In this vein, it is shown that a net positive charge associated with hexatic disclinations can be associated with the maximum curvature of dome-like structures in model organoids and in epithelial cell layers (**Rozman et al., 2020**; **Rozman et al., 2021**; **Hoffmann et al., 2022**). Moreover, in cellular monolayers, comet- and trefoil-shaped half-integer topological defects, corresponding to +1/2 and -1/2 charges, respectively, are prevalent (**Doostmohammadi et al., 2015**; **Doostmohammadi et al., 2016**). These are singular points in cellular alignment that mark the breakdown of orientational order (**de Genne and Prost, 1998**). Recent experiments on epithelial monolayers found a strong correlation between extrusion events and the position of a subset of +1/2 defects in addition to a relatively weaker correlation with -1/2 defects (**Saw et al., 2017**). These recently introduced purely mechanical routes to cell extrusion have opened the door to new questions on the nature of forces that are involved in eliminating cells from the monolayer and challenge the purely biological consensus that an extruding cell sends a signal to its neighbor that activates its elimination process (**Rosenblatt et al., 2001**). Nevertheless, it is not clear whether these different mechanisms are related, and whether, depending on the mechanical features of the cells, the cell layers actively switch between different routes to eliminate the unwanted cells. Since all the existing studies so far have only focused on effective two-dimensional models of the cell layers, fundamental questions about the three-dimensional phenomenon of cell extrusion and its connection to the interplay between cell-generated forces at the interface between cells and the substrate, with multicellular force transmission across the cell layer, remain unanswered.

In this article, we explore three-dimensional collective cell migration in cellular monolayers. Based on large-scale simulations, we examine (i) the underlying mechanisms responsible for cell extrusion, including any correlations with ±1/2 topological defects and fivefold disclinations, and (ii) the

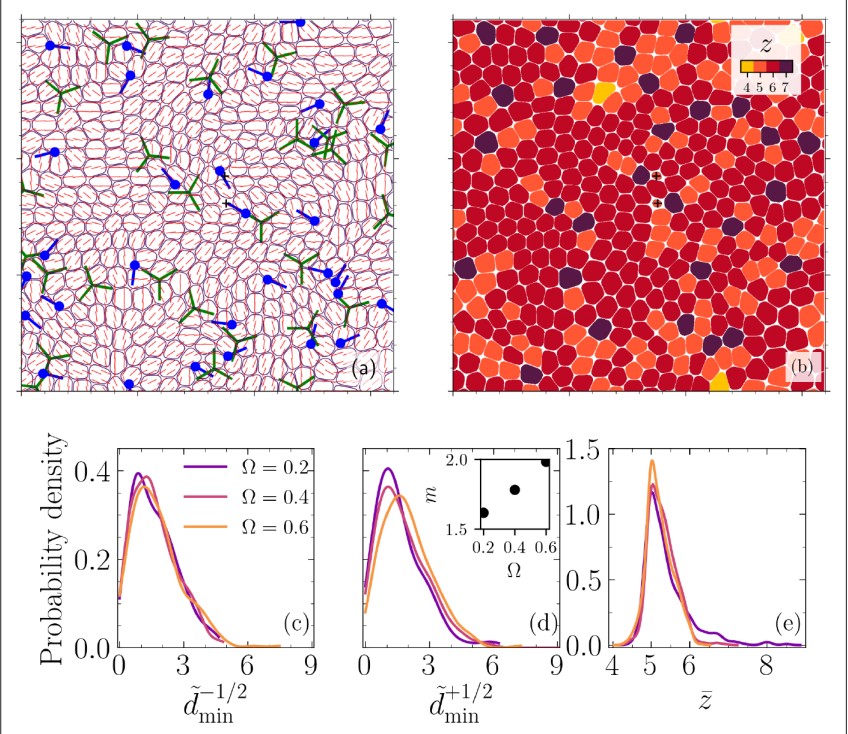

**Figure 2.** Nematic and hexatic disclinations govern cell extrusion. A representative analysis corresponding to the configuration shown in *Figure 1a* and projected into $xy-$ plane ($z = 0$, i.e., the basal side). (**a**) A coarse-grained director field with coarse-graining length of one cell size $\ell_{\text{dir.}} = R_0$ and +1/2 (filled circles with the line indicating orientation) and -1/2 (three connected lines with threefold symmetry) nematic defects. (**b**) Number of neighbors $z$ for each cell, including fivefold and sevenfold disclinations mapped into the monolayer. The symbol + denotes the center of mass for two extruding cells. (**c, d**) Probability densities of the normalized minimum distance between extruding cells and the nearest ±1/2 defect, $\tilde{d}_{\text{min}} = d_{\text{min}}/R_0$, for varying cell–cell to cell–substrate adhesion ratios $\Omega$ for (**c**) -1/2 and (**d**) +1/2 topological defects (inset: distribution mean $m = \langle \tilde{d}_{\text{min}}^{+1/2} \rangle$ vs. $\Omega$). (**e**) The probability density of average coordination number $\bar{z}$ for an extruding cell during $\tilde{t} = \left(t/\tau_0\right) \in [\tilde{t}_e - 2.5, \tilde{t}_e + 0.3125]$, where $\tilde{t}_e$ denotes extrusion time, $\tau_0 = \xi R_0/\alpha$ and for varying cell–substrate to cell–cell adhesion ratios $\Omega$. The data in (**c–e**) corresponds to four different realizations.

interplay of cell–cell and cell–substrate adhesion with extrusion events in cellular systems. Moreover, by mapping the full three-dimensional mechanical stress field across the entire monolayer, we identify localized stress concentration as the unifying factor that governs distinct topological routes to cell extrusion.

## Results and discussion

### Topological routes to cell extrusion: Nematic and hexatic disclinations

In the absence of self-propulsion forces, the initial configuration tends to equilibrate into a hexagonal lattice (see *Appendix 1—figure 11* in Appendix 1 for an example). As we introduce self-propulsion forces associated with front-rear cell polarity (see 'Materials and methods' for polarization dynamics), the system is pushed away from its equilibrium hexagonal configuration, resulting in defects manifested as fivefold and sevenfold disclinations, as shown in Figure 2b. *Figure 1a* shows a simulation snapshot with two extrusion events taking place. An extrusion event is detected if the vertical displacement of a cell, relative to other cells in the monolayer, exceeds $R_0/2$, where $R_0$ is the initial cell radius. *Figure 1b* displays the out-of-plane normalized velocity profile, $\vec{\tilde{v}}_z = \left(\vec{v}\left(\vec{x}\right) \cdot \vec{e}_z\right)/v_z^{\text{max}}$ where $v_z^{\text{max}}$ is the maximum value of the velocity component in $\vec{e}_z$ direction in the displayed cross-section of the monolayer, clearly marking the extruding cells as they get expelled from the monolayer and lose contact with the substrate.

In order to probe the possible mechanical routes to cell extrusion, we begin by characterizing topological defects in cell orientation field and disclinations in cellular arrangements. To this end, we first map the orientation field of the cells from the 2D projected cell shape profile on $xy-$ plane ($z = 0$, i.e., the basal side) and identify topological defects as the singularities in the orientation field. The results (example snapshot in *Figure 2a*) show the continuous emergence of half-integer ($\pm1/2$), nematic, topological defects that spontaneously nucleate in pairs and follow chaotic trajectories before annihilation (see *Appendix 1—figure 9* in Appendix 1 for energy spectra characterization). It is noteworthy that unlike previous studies of active nematic behavior in 2D cell layers (*Mueller et al., 2019*; *Wenzel and Voigt, 2021*), the nematic defects here emerge in the absence of any active dipolar stress or subcellular fields as the only active driving in these simulations is the polar force that the cells generate. Therefore, although the cells are endowed with polarity in terms of their self-propulsion, the emergent symmetry in terms of their orientational alignment is nematic, which is in line with experimental observations in cell monolayers (*Saw et al., 2017*; *Blanch-Mercader et al., 2018*), discrete models of self-propelled rods (*Bär et al., 2020*; *Meacock et al., 2021*), and recently proposed continuum model of polar active matter (*Amiri et al., 2022*).

Remarkably, in accordance with experimental observations (*Saw et al., 2017*), we find that the extrusion events can be correlated with the position of both +1/2 comet-shaped and -1/2 trefoil-shaped topological defects. To quantify this, *Figure 2c and d* display the probability density of the normalized minimum distance $\tilde{d}_{min}^{\pm1/2} = d_{min}^{\pm1/2}/R_0$ between an extruding cell and $\pm1/2$ topological defects in the interval $\tilde{t} \in [\tilde{t}_e - 5.625, \tilde{t}_e + 0.625]$, where $\tilde{t} = t/\tau_0$ is the normalized time, $\tau_0 = \xi R_0/\alpha$,

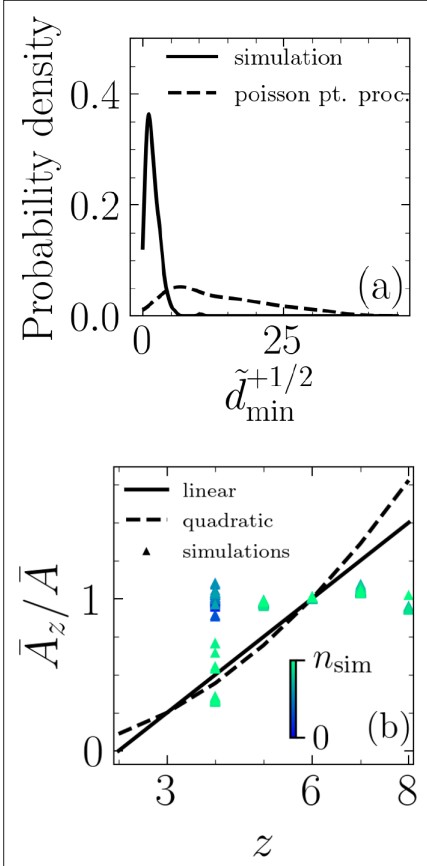

**Figure 3.** Topological, rather than geometrical, route to cell extrusion. (**a**) Probability density functions for normalized minimum distance between an extrusion event and a +1/2 defect, $\tilde{d}_{min}^{+1/2}$, based on simulation results and randomly generated through a Poisson point process and for $\Omega = 0.4$. (**b**) Comparison of Lewis's linear and quadratic relations with our simulations. $\bar{A}_z$ is the average area for cells with $z$ neighbors and $\bar{A}$ is the average area of all cells. The color bar indicates simulation time step, and the data correspond to the case $\omega_{cw} = 0.0025$ and $\Omega = 0.4$.

$\xi$ corresponds to cell–substrate friction, $\alpha$ denotes the strength of polarity force, and $\tilde{t}_e$ is the (normalized) extrusion time. This temporal window is chosen based on the first moment of a defect's lifetime distribution (see *Appendix 1—figure 5* in Appendix 1). The data in *Figure 2c and d* is based on four distinct realizations and for varying cell–substrate to cell–cell adhesion ratios, $\Omega = \omega_{cc}/\omega_{cw}$. For both defect types, the probability density peaks in the vicinity of the eliminated cell ($\approx 1.5R_0$), at a much smaller distance relative to a typical distance between two defects (see *Appendix 1—figure 7* in Appendix 1), and falls off to nearly zero for $d_{min}^{\pm1/2} 5R_0 (= 40)$. Furthermore, laser ablation experiments have established that an induced extrusion event does not favor the nucleation of a pair of nematic defects (*Saw et al., 2017*).

In a hypothesis-testing approach, we check whether these peaks in the minimum distance represent a correlation between extrusion events and nematic defects. To this end, we set out to falsify the hypothesis that the extrusion events are uncorrelated with the nematic defects. We utilize a Poisson point process to randomly generate positions for extrusion events and quantify the minimum distance between each event and the nearest half-integer nematic defect. For each simulation, we generate five different realizations for the extrusion events using a Poisson point process with the intensity set

equal to the number of extrusions in that particular simulation. The extrusion time is also a random variable described by a uniform distribution, $t_e \sim U(1, n_{\mathrm{sim}})$, where $n_{\mathrm{sim}} = 29,000$ is total number of time steps. As an example, *Figure 3a* shows probability density function for $\tilde{d}_{\mathrm{min}}^{+1/2}$ for $\Omega = 0.4$ using simulation data as well as data randomly generated with Poisson point process. Finally, a Kolmogorov–Smirnov (KS) test is used to measure if the two samples, one based on our simulations and one based on randomly generated extrusion events, belong to the same distribution. The results of the KS test reject this (see *Appendix 1—table 1* and *Appendix 1—figure 1* in Appendix 1) and thus falsify the hypothesis that simulation-based extrusion events are uncorrelated with the half-integer nematic defects.

Next, we explore the other possible mechanical route to cell extrusion based on the disclinations in cellular arrangement. To this end, we compute the coordination number of each cell based on their phase-field interactions and identify the fivefold and sevenfold disclinations (see *Figure 2b*). To quantify the relation between extrusion events and the disclinations, the probability density of the coordination number of an extruding cell ($\tilde{d}_{\mathrm{min}} = 0$) averaged over the time interval, $\tilde{t} \in [\tilde{t}_e - 5.625, \tilde{t}_e + 0.625]$, $\bar{z}$, for all the realizations is shown in *Figure 2e*, clearly exhibiting a sharp peak near $\bar{z} = 5$. The coordination number is determined based on the interactions of cells (see Appendix 1) and this property is independent of apical or basal considerations (*Kaliman et al., 2021*), unlike geometrical structures called scutoids that have been identified in curved epithelial tubes (*Gómez-Gálvez et al., 2018*). In our setup, the asymmetric interactions of cells with apical and basal sides are captured by varying the strength of cell–substrate adhesion. In our simulations, increasing cell–substrate adhesion leads to lower extrusion events (see *Appendix 1—figure 8* in Appendix 1).

Thus far, our results suggest topological rather than geometrical routes to cell extrusion. To probe the role of geometrical constraints further, we investigate the existence of any correlation between cell area and its number of neighbors. The best known such a correlation – for cellular matter with no gaps between them, that is, confluent state – is a linear one and it is due to *Lewis, 1928* with other types of relations, for example, quadratic, proposed since his work (*Kokic et al., 2019*). We compare our simulation results against both the linear ($\bar{A}_z/\bar{A} = (z - 2)/4$ where $\bar{A}_z$ is the average area of cell with $z$ neighbors and $\bar{A}$ is the average area of all cells) and quadratic relations ($\bar{A}_z/\bar{A} = (z/6)^2$) and find the agreement poor, as shown for the case of $\omega_{\mathrm{cw}} = 0.0025$ and $\Omega = 0.4$ in *Figure 3b* (see also *Appendix 1—figures 3 and 4* in Appendix 1). While in our simulations the cell monolayers are not always confluent due to the extrusion events, other studies with confluent cellular layers have also found such relations to not be valid (*Kim et al., 2014*; *Wenzel et al., 2019*). In our simulations, the projected area of an extruding cell decreases prior to extrusion, but the number of interacting neighbors generally does not change in that time frame (see *Appendix 1—figure 4* in Appendix 1). Together, these results suggest mechanical rather than geometrical routes to cell extrusion. Specifically, in our approach cell extrusion emerges as a consequence of cells pushing and pulling on their neighbors due to their intrinsic activity. This contrasts with inherently threshold-based vertex models (see, e.g., *Okuda and Fujimoto, 2020*) for both cellular rearrangements (T1 transitions) and extrusions (T2 transitions).

## Mechanical stress localization unifies distinct topological routes to cell extrusion

The correlation between disclinations and extrusion events is also related to the mechanical stress localization at the fivefold disclinations: The occurrence of disclinations in a flat surface produces local stress concentration (*Irvine et al., 2010*). Generally, it is energetically favorable to bend a flat surface, rather than to compress or to stretch it (*Landau et al., 1986*). Thus, the local stress concentration can lead to a fivefold (positive Gaussian curvature) or a sevenfold (negative Gaussian curvature) disclination (*Seung and Nelson, 1988*; *Guitter and Kardar, 1990*). In our set-up and given that we consider a rigid substrate, fivefold disclinations are much more likely to provide relief for the high local stress concentration. This can change if the rigidity of substrate is relaxed or extrusion in three-dimensional spheroids are considered. Since we conjecture that both topological defect- and disclination-mediated extrusion mechanisms are closely linked with stress localization, we characterize the in-plane and out-of-plane stresses associated with the simulated monolayer. We compute a coarse-grained stress field (*Christoffersen et al., 1981*; *Li et al., 2022*), $\sigma_{ij} = (1/(2V_{\mathrm{cg}})) \sum_{m \in V_{\mathrm{cg}}} \left( \vec{T}_i(\vec{x}_m) \otimes \vec{e}_j^n + \vec{T}_j(\vec{x}_m) \otimes \vec{e}_i^n \right)$,

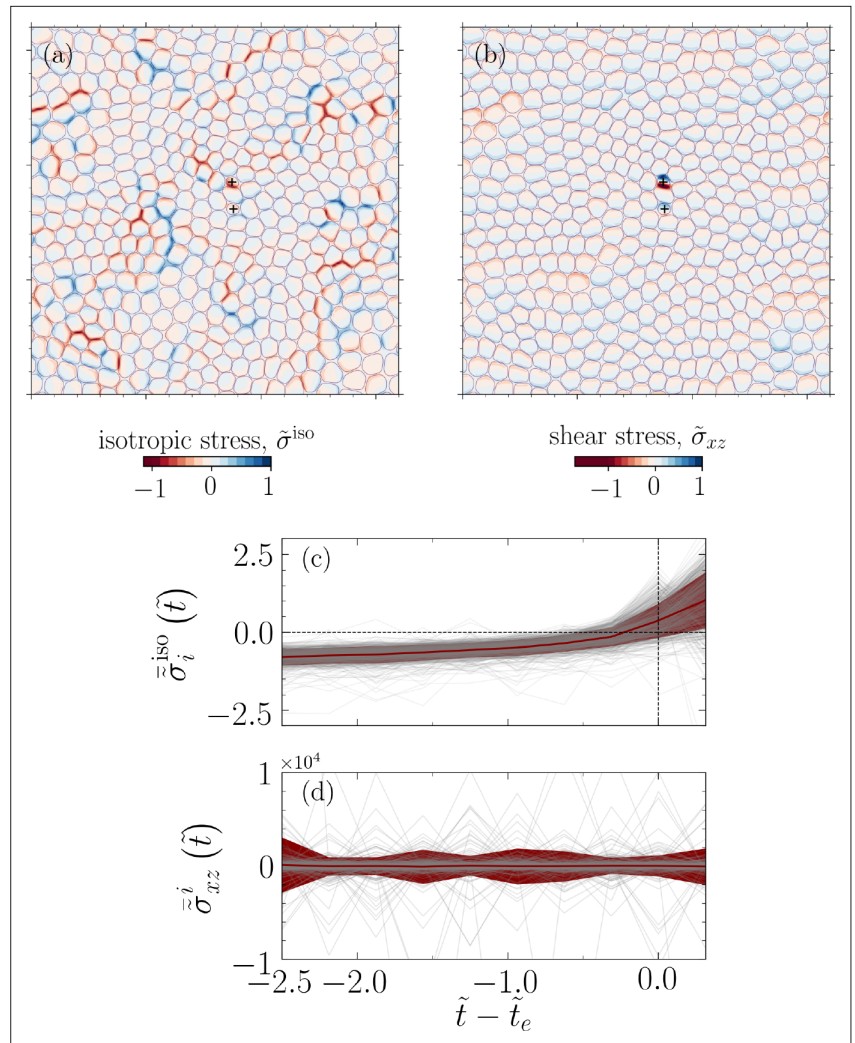

**Figure 4.** Temporal build-up of mechanical stress before extrusion events. A representative analysis corresponding to the configuration shown in *Figure 1a* and projected into $xy-$ plane ($z = 0$, i.e., the basal side). (a) Normalized isotropic stress field $\tilde{\sigma}^{\mathrm{iso}}(\vec{x}) = \sigma^{\mathrm{iso}}(\vec{x})/\sigma^{\mathrm{iso}}_{\mathrm{max}}$, where $\sigma^{\mathrm{iso}}_{\mathrm{max}}$ is the maximum value of the isotropic stress field, and (b) normalized shear stress field, $\tilde{\sigma}_{\mathrm{xz}}(\vec{x}) = \sigma_{\mathrm{xz}}(\vec{x})/\sigma^{\mathrm{max}}_{\mathrm{xz}}$, where $\sigma^{\mathrm{max}}_{\mathrm{xz}}$ is the maximum value of $\sigma_{\mathrm{xz}}(\vec{x})$ field. The symbol + denotes the center of mass for two extruding cells. (c) Cell (spatially) averaged normalized isotropic stress $\bar{\bar{\sigma}}^{\mathrm{iso}}_i(\tilde{t}) = \langle\sigma^{\mathrm{iso}}(\vec{x},\tilde{t})\rangle_{\vec{x}\in\mathcal{R}_i}/\langle\sigma^{\mathrm{iso}}(\vec{x},\tilde{t})\rangle_{\vec{x}\in\mathcal{R}}$ and (d) shear stress $\bar{\bar{\sigma}}^i_{\mathrm{xz}}(\tilde{t}) = \langle\sigma_{\mathrm{xz}}(\vec{x},\tilde{t})\rangle_{\vec{x}\in\mathcal{R}_i}/\langle\sigma_{\mathrm{xz}}(\vec{x},\tilde{t})\rangle_{\vec{x}\in\mathcal{R}}$ for an extruding cell $i$ during $\tilde{t} = (t/\tau_0) \in [\tilde{t}_e - 2.5, \tilde{t}_e + 0.3125]$, where $\tilde{t}_e$ denotes extrusion time and $\tau_0 = \xi R_0/\alpha$. The data shown in (c, d) correspond to all the considered parameters for cell–substrate ($\omega_{\mathrm{cw}}$) and relative cell–cell adhesions ($\Omega$) and for four distinct realizations. Each gray line in the background represents an extruding cell, and the red line shows the mean and the standard deviation of the normalized stresses.

where $\vec{x}_0$ represents the center of the coarse-grained volume, $V_{\mathrm{cg}} = \ell^3_{\mathrm{stress}}$, corresponding to coarse-grained length $\ell_{\mathrm{stress}}$ and unit vector $\vec{e}^{\eta}_i = (\vec{x}_0 - \vec{x}_m)/|\vec{x}_0 - \vec{x}_m|$. Herein, the stress fields are computed using $\ell_{\mathrm{stress}} = R_0/4$.

For the example simulation snapshot displayed in *Figure 1a*, at the onset of two extrusion events, we visualize normalized isotropic stress $\tilde{\sigma}^{\mathrm{iso}}(\vec{x}) = \sigma^{\mathrm{iso}}(\vec{x})/\sigma^{\mathrm{iso}}_{\mathrm{max}}$ and out-of-plane shear $\tilde{\sigma}_{\mathrm{xz}}(\vec{x}) = \sigma_{\mathrm{xz}}(\vec{x})/\sigma^{\mathrm{max}}_{\mathrm{xz}}$, where $\sigma^{\mathrm{iso}}_{\mathrm{max}}$ and $\sigma^{\mathrm{max}}_{\mathrm{xz}}$ are the maximum values in their corresponding fields (see *Figure 4a and b*). We observe a high, out-of-plane, shear stress concentration (*Figure 4b*) as well as tensile and compressive stress pathways (*Figure 4a*) reminiscent of force chains in granular systems (*Majmudar and Behringer, 2005*).

*Figure 4c* shows the evolution of spatially averaged normalized isotropic stress for extruding cell $i$, $\bar{\bar{\sigma}}^{\mathrm{iso}}_i(\tilde{t}) = \langle\sigma^{\mathrm{iso}}(\vec{x},\tilde{t})\rangle_{\vec{x}\in\mathcal{R}_i}/\langle\sigma^{\mathrm{iso}}(\vec{x},\tilde{t})\rangle_{\vec{x}\in\mathcal{R}}$, demonstrating a clear stress build-up, followed by a

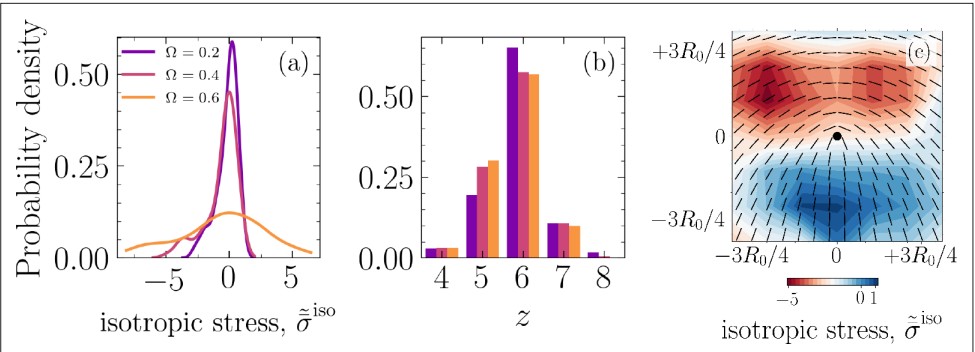

**Figure 5.** Spatial localization of mechanical stress leading to extrusion events. (**a**) Probability density function for the normalized, ensemble average of isotropic stress field, $\tilde{\bar{\sigma}}^{\text{iso}}$, projected into $xy$-plane with $z = 0$, that is, the basal side, around +1/2 defects for various cell–cell to cell–substrate adhesion ratios $\Omega$ (colors correspond to legend in [**a**]). (**b**) Probability density function for the number of neighbors $z$ and various $\Omega$, for all cells and simulation time steps. (**c**) Normalized, ensemble average of isotropic stress field, $\tilde{\bar{\sigma}}^{\text{iso}}(\vec{x}) = \bar{\sigma}^{\text{iso}}(\vec{x})/\bar{\sigma}^{\text{iso}}_{\max}$, with $\bar{\sigma}^{\text{iso}}$ representing the average field during defect life time, for nucleated defects, and $\bar{\sigma}^{\text{iso}}_{\max}$ is the maximum of the average field, for the case $\omega_{\text{cw}} = 0.0025$ and $\Omega = 0.4$.

drop near $\tilde{t} - \tilde{t}_e = 0$, as a cell detaches the substrate and loses contact with other cells, where $\tilde{t}_e$ is detected by our stress-independent criterion, $\mathcal{R} = \bigcup_{i=1}^{N} \mathcal{R}_i$ and $\mathcal{R}_i$ is the domain associated with cell $i$, $\mathcal{R}_i := \{\vec{x} | \phi_i(\vec{x}) \geq 0.5\}$.

Similarly, **Figure 4d** displays the spatially averaged normalized out-of-plane shear stress, $\tilde{\bar{\sigma}}^i_{xz}(\tilde{t}) = \langle \sigma_{xz}(\vec{x}, \tilde{t}) \rangle_{\vec{x} \in \mathcal{R}_i} / \langle \sigma_{xz}(\vec{x}, \tilde{t}) \rangle_{\vec{x} \in \mathcal{R}}$, prior to a cell extrusion and for all extrusion events in our simulations, that is, nine cases and four realizations for each case. The shear stress prior to extrusion exhibits oscillations with large magnitudes relative to the mean field, a stark departure from their non-extruding counterparts (see **Appendix 1—figure 4** in Appendix 1). This may indicate a hindrance to cell movement as we explore further next.

Interestingly, the association of cell extrusion events with regions of high out-of-plane shear stress has parallels with the phenomenon of *plithotaxis*, where it was shown that cells collectively migrate along the orientation of the minimal in-plane intercellular shear stress (**Tambe et al., 2011**). In this context, based on the association of cell extrusion events with regions of high out-of-plane shear stress, we conjecture that high shear stress concentration hinders collective cell migration with cell extrusion providing a mechanism to re-establish the status quo. This is also consistent with the observation we made earlier about large oscillations in $\tilde{\bar{\sigma}}^i_{xz}$ prior to an extrusion event for all extruding cells (**Figure 4d**).

## Shifting tendencies towards extrusion at nematic and hexatic disclinations

The results so far clearly demonstrate the existence of mechanical routes for cell removal that are associated with nematic and hexatic disclinations and are governed by the in-plane and out-of-plane mechanical stress patterns in the cell assembly. The relative strength of cell–cell to cell–substrate adhesion, $\Omega$, further alters the likelihood of an extrusion event being associated with a +1/2 defect or a fivefold disclination. This is clearly observed in **Figure 2d**, which shows the first moment of the distribution for $\tilde{d}^{+1/2}_{\min}$, $m = \langle \tilde{d}^{+1/2}_{\min} \rangle$, increases with $\Omega = \omega_{\text{cc}}/\omega_{\text{cw}}$ (see inset) while the peak of the probability density decreases with increasing $\Omega$. At the same time, the probability of an extrusion occurring at a fivefold disclination increases with increasing $\Omega$, as displayed in **Figure 2e**. However, nematic and hexatic order parameters do not show any clear trends with $\Omega$ (see **Appendix 1—figure 10** in Appendix 1). To better understand this tendency, we characterize the average isotropic stress fields around a +1/2 defect. This involves tracking each +1/2 defect starting from its nucleation and mapping the isotropic stress field, for each time step during the defect's lifetime, in a square domain of size $L \approx 1.5R_0$ centered on the defect location and accounting for its orientation, where $L$ is chosen based on the peak in $\tilde{d}^{+1/2}_{\min}$ (see **Figure 2d**). An example for the normalized average isotropic stress field corresponding to $\omega_{\text{cw}} = 0.0025$ and $\Omega = 0.4$ is shown in **Figure 5c**, where $\tilde{\bar{\sigma}}^{\text{iso}}(\vec{x}) = \bar{\sigma}^{\text{iso}}(\vec{x})/\bar{\sigma}^{\text{iso}}_{\max}$,

with $\bar{\sigma}^{\mathrm{iso}}$ representing the average field during defect lifetime, for all nucleated defects, and $\bar{\sigma}^{\mathrm{iso}}_{\mathrm{max}}$ is the maximum of the average field. This is in agreement with experimental measurements on epithelial monolayers (*Saw et al., 2017*; *Balasubramaniam et al., 2021*), with a compressive stress region near the head of the defect and a tensile region near the tail. Interestingly, there is an asymmetry in the intensity of stress in the compressive region at the head of the comet as opposed to the tensile region at the tail ($\approx 5\times$ higher). To expand on this observation, we focus on the probability density function for $\tilde{\sigma}^{\mathrm{iso}}$ and various $\Omega$. Remarkably, as shown in *Figure 5a*, with increasing $\Omega$, the peak of the probability density function decreases and the shoulders become wider, that is, the stress localization becomes more spread. At the same time, it is worth noting that the probability for the occurrence of a fivefold disclination increases as $\Omega$ is increased, as shown in *Figure 5b*, while no clear trend is observed for the density of half-integer defects (see *Appendix 1—figure 6* in Appendix 1). Therefore, the more spread localized stress is more likely to only clear the lower energetic barrier associated with buckling of a fivefold disclination (*Seung and Nelson, 1988*; *Guitter and Kardar, 1990*) – forming a positive Gaussian curvature – as opposed to cells with six neighbors. Furthermore, for a single disclination, this energy is higher for a sevenfold disclination (*Deem and Nelson, 1996*) and in our case the rigid substrate defies any attempts by a sevenfold disclination to buckle and form a negative Gaussian curvature. Together, these results provide a potential explanation for why as $\Omega$ is increased, cells collectively have a tendency towards leveraging fivefold disclinations instead of +1/2 defects for extruding an unwanted cell.

Furthermore, one may naively think that only the distance between a half-integer nematic defect and an extrusion site is of importance. Such a view implicitly assumes the statistics we have presented (e.g., $\tilde{d}^{\pm 1/2}_{\mathrm{min}}$, $\tilde{\sigma}^{\mathrm{iso}}$) correspond to independent events, disregarding the highly heterogeneous nature of such a complex, active system. This heterogeneous nature manifests in stress fields, as shown in *Figure 5a and c* for the normalized ensemble average around a +1/2 defect. Therefore, the distance between a defect and an extrusion site, the intensity and the extent of the stress fields around that defect all play a role and are embedded in the statistics that we present in this work. In the future, it can be illuminating to study the effect of heterogeneity in the apical–basal mechanical response due to different mechanical properties and/or the nature of activity.

## Conclusions

Our study presents a three-dimensional model of the collective migration-mediated cell elimination. Importantly, this framework allows for cell–substrate and cell–cell adhesion forces to be tuned independently. Our findings indeed suggest that varying the relative strength of cell–cell and cell–substrate adhesion can allow cells to switch between distinct mechanical pathways – leveraging defects in nematic and hexatic phases – to eliminate unwanted cells through: (i) cell extrusion at ±1/2 topological defects in the cell orientation field, consistent with experimental observations (*Saw et al., 2017*); and (ii) cell extrusion at fivefold disclinations in cell arrangement, where our results show a direct role of these disclinations in extruding the cells. Focusing on the extruded cells, the results demonstrate that increasing relative cell–cell adhesion increases the probability of an extruded cell being a fivefold disclination while weakening the correlation with +1/2 topological defects. This seems to emerge with a confluence of factors at play: (i) higher likelihood for a cell to be a fivefold disclination as $\Omega = \omega_{\mathrm{cc}}/\omega_{\mathrm{cw}}$ is increased, (ii) more spread stress concentration around a +1/2 defect with increasing $\Omega$, and (iii) a higher likelihood for such a diffused local stress field to only reach the lower energy barrier associated with buckling a fivefold disclination (forming a positive Gaussian curvature) as opposed to cells with six neighbors as well as sevenfold disclinations. In the latter case, in addition to higher energy barrier, the rigid substrate denies a sevenfold disclination to create any negative Gaussian curvatures.

Additionally, the presented framework provides access to the local stress field, including the out-of-plane shear components. Access to this information led us to conjecture that high shear stress concentration frustrates collective cell migration with cell extrusion providing a pathway to re-establish the status quo. We expect these results to trigger further experimental studies of the mechanical routes to live cell elimination and probing the impact of tuning cell–cell and cell–substrate interactions, for example, by molecular perturbations of E-cadherin adhesion complexes between the cells and/or focal adhesion between cells and substrate, as performed recently in the context of topological defect motion in cell monolayers (*Balasubramaniam et al., 2021*). In this study, we intentionally narrowed our focus to the interplay of cell–cell and cell–substrate adhesion, without accounting for

cell proliferation. In its absence, simulations with high extrusion events may lose confluency. However, the identified mechanical routes to extrusion prevail in cases with both high and low number of extrusions, where confluency is maintained.

Finally, we anticipate that this modeling framework opens the door to several interesting and unresolved problems in studying three-dimensional features of cell layers. In particular, the mechanics can be coupled with biochemistry to study a wider range of mechanisms that affect live cell elimination. Additionally, using our framework the substrate rigidity can be relaxed in the future studies to further disentangle the impacts of cell–substrate adhesion from substrate deformation due to cell generated forces. Similarly, three-dimensional geometries, such as spheroids or cysts, can be examined. The links between collective cell migration and granular physics, in terms of force chains and liquid-to-solid transition, as well as probing the impact of three-dimensionality and out-of-plane deformations on these processes, are exciting directions for future studies. Lastly, the coexistence of nematic and hexatic phases, their potential interactions, and their interplay with curved surfaces are promising avenues for extending the work presented here.

## Materials and methods

We consider a cellular monolayer consisting of $N = 400$ cells on a substrate with its surface normal $\vec{e}_n (= \vec{e}_z) = \vec{e}_x \times \vec{e}_y$ and periodic boundaries in both $\vec{e}_x$ and $\vec{e}_y$, where $(\vec{e}_x, \vec{e}_y, \vec{e}_z)$ constitute the global orthonormal basis (*Figure 1*). Cells are initiated on a two-dimensional simple cubic lattice and inside a cuboid of size $L_x = L_y = 320$, $L_z = 64$, grid size $a_0 = 1$ and with radius $R_0 = 8$. The cell–cell and cell–substrate interactions have contributions from both adhesion and repulsion, in addition to self-propulsion forces associated with cell polarity. To this end, each cell $i$ is modeled as an active deformable droplet in three-dimensions using a phase-field, $\phi_i = \phi_i(\vec{x})$. The interior and exterior of cell $i$ corresponds to $\phi_i = 1$ and $\phi_i = 0$, respectively, with a diffuse interface of length $\lambda$ connecting the two regions and the midpoint, $\phi_i = 0.5$, delineating the cell boundary. A three-dimensional extension of the 2D free energy functional (*Palmieri et al., 2015*; *Aranson, 2016*; *Camley and Rappel, 2017*; *Mueller et al., 2019*) is considered with additional contributions to account for cell–cell and cell–substrate adhesions:

$$\mathcal{F} = \sum_i^N \frac{\gamma}{\lambda} \int d\vec{x} \{4\phi_i^2 (1 - \phi_i)^2 + \lambda^2 (\vec{\nabla}\phi_i)^2\} +$$

$$\sum_i^N \mu \left(1 - \frac{1}{V_0} \int d\vec{x}\phi_i^2\right)^2 + \sum_i^N \sum_{j \neq i} \frac{\kappa_{cc}}{\lambda} \int d\vec{x}\phi_i^2 \phi_j^2 +$$

$$\sum_i^N \sum_{j \neq i} \frac{\omega_{cc}}{\lambda^2} \int d\vec{x} \vec{\nabla}\phi_i \cdot \vec{\nabla}\phi_j + \sum_i^N \frac{\kappa_{cw}}{\lambda} \int d\vec{x}\phi_i^2 \phi_w^2 +$$

$$\sum_i^N \frac{\omega_{cw}}{\lambda^2} \int d\vec{x} \vec{\nabla}\phi_i \cdot \vec{\nabla}\phi_w, \tag{1}$$

where $\mathcal{F}$ contains a contribution due to the Cahn–Hilliard free energy (*Cahn and Hilliard, 1958*), which stabilizes the cell interface, followed by a soft constraint for cell volume around $V_0 \left(= (4/3)\pi R_0^3\right)$, such that cells – each initiated with radius $R_0$ – are compressible. Additionally, $\kappa$ and $\omega$ capture repulsion and adhesion between cell–cell (subscript $cc$) and cell–substrate (subscript $cw$), respectively. Moreover, $\gamma$ sets the cell stiffness and $\mu$ captures cell compressibility and $\phi_w$ denotes a static phase-field representing the substrate. This approach resolves the cellular interfaces and provides access to intercellular forces. The dynamics for field $\omega$ can be defined as:

$$\partial_t \phi_i + \vec{v}_i \cdot \vec{\nabla}\phi_i = -\frac{\delta\mathcal{F}}{\delta\phi_i}, \qquad i = 1, ..., N, \tag{2}$$

where $\mathcal{F}$ is defined in *Equation (1)*, and $\vec{v}_i$ is the total velocity of cell $i$. To resolve the forces generated at the cellular interfaces, we consider the following over-damped dynamics for cells:

$$\vec{t}_i = \xi\vec{v}_i - \vec{F}_i^{\text{sp}} = \int d\vec{x}\phi_i \vec{\nabla} \cdot \mathbf{\Pi}^{\text{int}}, \tag{3}$$

where $\vec{t}_i$ denotes traction as defined for Bayesian Inversion Stress Microscopy in *Saw et al., 2017*, $\xi$ is substrate friction, and $\vec{F}_i^{\text{sp}} = \alpha\vec{p}_i$ represents self-propulsion forces due to polarity, constantly pushing the system out-of-equilibrium. In this vein, $\alpha$ characterizes the strength of polarity force and $\mathbf{\Pi}^{\text{int}} = \left(\sum_i - (\delta\mathcal{F}/\delta\phi_i)\right)\mathbf{1}$. While only passive interactions are considered here, active nematic

interactions can be readily incorporated in this framework (*Mueller et al., 2019*; *Balasubramaniam et al., 2021*). To complete the model, the dynamics of front-rear cell polarity is introduced based on contact inhibition of locomotion (CIL) (*Abercrombie and Heaysman, 1954*; *Abercrombie, 1979*) by aligning the polarity of the cell to the direction of the total interaction force acting on the cell (*Smeets et al., 2016*; *Peyret et al., 2019*). As such, the polarization dynamics is given by

$$\partial_t \theta_i = -J|\vec{t}_i|\Delta\theta_i + D_r\eta, \tag{4}$$

where $\theta_i \in [-\pi, \pi]$ is the angle associated with polarity vector, $\vec{p}_i = (\cos\theta_i, \sin\theta_i, 0)$, and $\eta$ is the Gaussian white noise with zero mean, unit variance, $D_r$ is rotational diffusivity, $\Delta\theta_i$ is the angle between $\vec{p}_i$ and $\vec{t}_i$, and positive constant $J$ sets the alignment time scale. It is worth noting that the self-propulsion forces, $\vec{F}_i^{\mathrm{sp}}$, associated with cell polarity, $\vec{p}_i$, act in-plane but can induce out-of-plane components in force and velocity fields as a cell described by $\phi_i(\vec{x})$ deforms in three-dimensions (see *Equation 3*).

We perform large-scale simulations with a focus on the interplay of cell–cell and cell–substrate adhesion strengths and its impact on cell expulsion from the monolayer. To this end, we set the cell–substrate adhesion strength $\omega_{\mathrm{cw}} \in \{0.0015, 0.002, 0.0025\}$ and vary the cell–substrate to cell–cell adhesion ratio in the range $\Omega = \omega_{\mathrm{cc}}/\omega_{\mathrm{cw}} \in \{0.2, 0.4, 0.6\}$. For each case in this study (total of nine), we simulate four distinct realizations with a total of $n_{\mathrm{sim}} = 29,000$ time steps. All results are reported in dimensionless units, introduced throughout the text, and the simulation parameters are chosen within the range that was previously shown to reproduce defect flow fields in epithelial layers (*Balasubramaniam et al., 2021*; see Appendix 1).

## Acknowledgements

SM is grateful for the generous support of the Rosenfeld Foundation fellowship at the Niels Bohr Institute, University of Copenhagen. SM, GR, and JA acknowledge support for this research provided by US ARO funding through the Multidisciplinary University Research Initiative (MURI) grant no. W911NF-19-1-0245. AD acknowledges funding from the Novo Nordisk Foundation (grant no. NNF18SA0035142 and NERD grant No. NNF21OC0068687), Villum Fonden grant no. 29476, and the European Union via the ERC-Starting Grant PhysCoMeT. The authors would like to thank Dr. Lakshmi Balasubramaniam and Prof. Benoît Ladoux (Institut Jacques Monod, University Paris City), Guanming Zhang and Prof. Julia M Yeomans (The Rudolf Peierls Centre for Theoretical Physics, University of Oxford), Prof. Jörn Dunkel (Mathematics Department, MIT), and Prof. M Cristina Marchetti (Department of Physics, University of California Santa Barbara) for helpful discussions. The authors are also grateful for the comments and the feedback provided by the anonymous reviewers.

## Additional information

### Funding

| Funder | Grant reference number | Author |
| --- | --- | --- |
| Multidisciplinary University Research Initiative | W911NF-19-1-0245 | Guruswami Ravichandran José Andrade |
| Novo Nordisk Fonden | NNF18SA0035142 | Amin Doostmohammadi |
| Villum Fonden | 29476 | Amin Doostmohammadi |
| Novo Nordisk Foundation | NNF21OC0068687 | Amin Doostmohammadi |

The funders had no role in study design, data collection and interpretation, or the decision to submit the work for publication.

### Author contributions

Siavash Monfared, Conceptualization, Software, Formal analysis, Investigation, Visualization, Methodology, Writing – original draft, Writing – review and editing; Guruswami Ravichandran, José Andrade, Conceptualization, Resources, Supervision, Funding acquisition, Investigation, Writing – review and

editing; Amin Doostmohammadi, Conceptualization, Resources, Supervision, Funding acquisition, Investigation, Methodology, Writing – review and editing

**Author ORCIDs**
Siavash Monfared (iD) http://orcid.org/0000-0002-7629-7977

**Decision letter and Author response**
Decision letter https://doi.org/10.7554/eLife.82435.sa1
Author response https://doi.org/10.7554/eLife.82435.sa2

## Additional files

**Supplementary files**
• MDAR checklist

**Data availability**
The current manuscript is a computational study, so no data have been generated for this manuscript. Modelling code is uploaded on the first author's GitHub page (https://github.com/siavashmonfared/siavashmonfared.github.io copy archived at *Monfared, 2023*).

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

## Appendix 1

## Kolmogorov–Smirnov test for correlation between extrusion events and nematic defects

We use a hypothesis-testing approach to explore the existence of a correlation between the extrusion events in our simulations and nucleation of nematic topological defects. Specifically, we hypothesize that the extrusion events are uncorrelated with the topological defects. To falsify this, we use a Poisson point process to randomly generate extrusion events and quantify the minimum distance, $\bar{d}_{\min} = d_{\min}/R_0$ between the extrusion location and the nearest half-integer topological defects. To this end, for each simulation, we generate five realization of extrusion events using a Poisson point process with intensity set equal to the number of extrusions in that particular simulation. Furthermore, we assign an extrusion time, $t_e$, using a uniform distribution $t_e \sim U\left(1, n_{\text{sim}}\right)$. Then, we quantify the minimum distance $\bar{d}_{\min}$ as we have done for our simulations. The results are shown in **Appendix 1—figure 1**. To falsify the stated hypothesis, we use a Kolmogorov–Smirnov (KS) test to measure if the two samples, one based in our simulations and one based on randomly generated extrusion events belong to the same distribution. As shown in **Appendix 1—table 1**, the results of the KS test falsify this hypothesis, that is, simulation based extrusion events are uncorrelated with the topological defects.

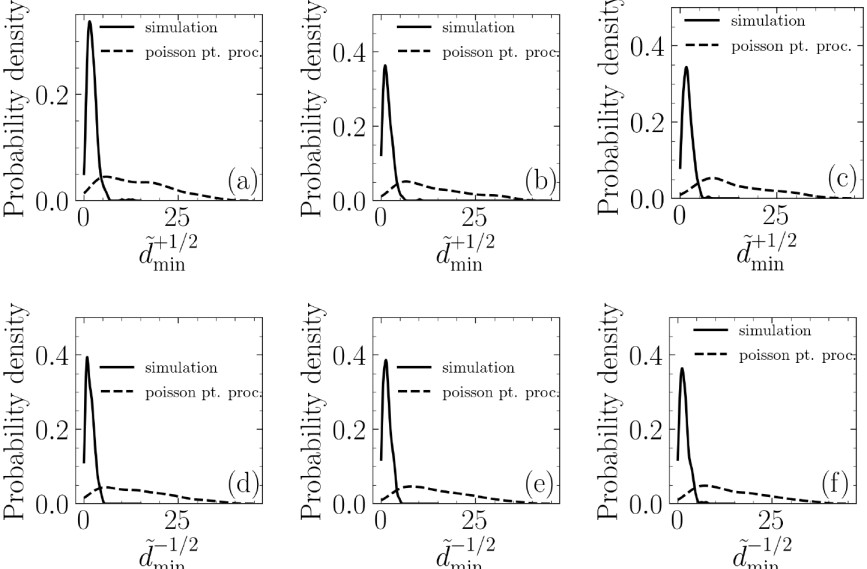

**Appendix 1—figure 1.** Establishing the correlation between extrusion events and half-integer nematic defects via hypothesis testing. Probability density of the normalized minimum distance, $\bar{d}_{\min}^{+1/2}$ between an extrusion event and the nearest +1/2 topological defect (**a–c**) for $\Omega = 0.2$ (**a**), $\Omega = 0.4$ (**b**), and $\Omega = 0.6$ (**c**). Probability density of the normalized minimum distance, $\bar{d}_{\min}^{-1/2}$ between an extrusion event and the nearest -1/2 topological defect (**d–f**) for $\Omega = 0.2$ (**d**), $\Omega = 0.4$ (**e**), and $\Omega = 0.6$ (**f**).

**Appendix 1—table 1.** Results of the Kolmogorov–Smirnov (KS) tests for various cell–cell to cell–substrate adhesion ratio ($\Omega = \omega_{\text{cc}}/\omega_{\text{cw}}$) and for half-integer topological defects.

Both statistics and p-value are KS test results and $n$ corresponds to the number of samples in each distribution, for both simulations and the extrusion events generated through a Poisson point process.

| Probability density | statistics | p-Value | $n$(simulations) | $n$(randomly generated) |
|---|---|---|---|---|
| $\Omega = 0.2(+1/2)$ | 0.8221 | $1.95 \times 10^{-154}$ | 426 | 402 |
| $\Omega = 0.4(+1/2)$ | 0.228 | $2.22 \times 10^{-15}$ | 648 | 605 |

*Appendix 1—table 1 Continued on next page*

*Appendix 1—table 1 Continued*

| Probability density | statistics | p-Value | $n$(simulations) | $n$(randomly generated) |
|---|---|---|---|---|
| $\Omega = 0.6(+1/2)$ | 0.827 | $5.12 \times 10^{-212}$ | 551 | 570 |
| $\Omega = 0.2(-1/2)$ | 0.802 | $1.22 \times 10^{-15}$ | 426 | 423 |
| $\Omega = 0.4(-1/2)$ | 0.840 | $8.17 \times 10^{-260}$ | 648 | 660 |
| $\Omega = 0.6(-1/2)$ | 0.824 | $2.84 \times 10^{-198}$ | 551 | 507 |

## Simulation parameters

We perform large-scale simulations with a focus on the interplay of cell–cell and cell–substrate adhesion strengths on collective cell migration and its impact on cell expulsion from the monolayer. Following *Mueller et al., 2019*, the space and time discretization in our simulations are based on the average radius of MDCK cells, $\sim 5\mu m$, velocity $\sim 20\mu m/h$, and average pressure of ~100 Pa, measured experimentally in MDCK monolayers (*Saw et al., 2017*), corresponding to $\Delta x \sim 0.5\mu m$, $\Delta t \sim 0.1s$, and $\Delta F \sim 1.5$ nN for force. In this study, we set the cell–substrate adhesion strength $\omega_{cw} \in \{0.0015, 0.002, 0.0025\}$ and vary the cell–substrate to cell–cell adhesion ratio in the range $\Omega = \omega_{cc}/\omega_{cw} \in \{0.2, 0.4, 0.6\}$. Based on previous experimental and theoretical studies (*Mueller et al., 2019*; *Peyret et al., 2019*; *Balasubramaniam et al., 2021*), the other simulation parameters are $\kappa_{cc} = 0.5$, $\kappa_{cw} = 0.15$, $\xi = 1$, $\alpha = 0.05$, $\lambda = 3$, $\mu = 45$, $D_r = 0.01$, and $J = 0.005$, unless stated otherwise.

## Lewis's empirical relation

The empirical relationship proposed by *Lewis, 1928* is generally valid for cellular matter that fill in the space without gaps. In our simulations, we can lose confluency due to cellular extrusions. Furthermore, even in the case of a confluent cellular layer, that is, no gaps between cells, Lewis's law fails to capture the correlation between normalized area and number of neighbors (see *Kim et al., 2014*; *Wenzel et al., 2019*). However, we still investigated the existence of such correlation in our simulations. To this end, we used the following linear and quadratic relationships (*Kokic et al., 2019*):

$$\frac{\bar{A}_z}{\bar{A}} = \frac{z-2}{4} \tag{5}$$

$$\frac{\bar{A}_z}{\bar{A}} = \left(\frac{z}{6}\right)^2 \tag{6}$$

where $\bar{A}_z$ is the average area of cells with $z$ neighbors and $\bar{A}$ is the average of area of the cells in the monolayer. As shown in *Appendix 1—figure 2*, the agreement is poor. Furthermore, while the projected area of an extruding cell decreases prior to extrusion, the number of neighbors its interacting with remains generally unchanged. This is shown in *Appendix 1—figure 2*.

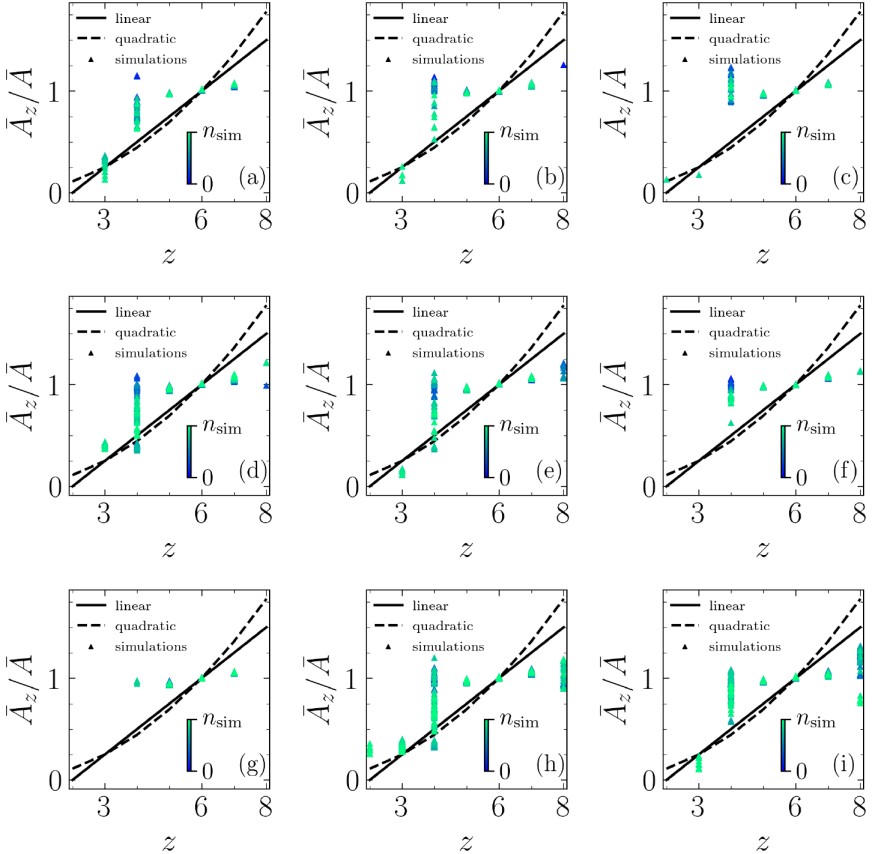

**Appendix 1—figure 2.** Lewis's relationship for extruding cells. (**a–i**) Lewis's relationship compared with our simulations for $\omega_{\text{cw}} = 0.0015$ and $\Omega = 0.2$ (**a**), $\Omega = 0.4$ (**b**), and $\Omega = 0.6$ (**c**); $\omega_{\text{cw}} = 0.002$ and $\Omega = 0.2$ (**d**), $\Omega = 0.4$ (**e**), and $\Omega = 0.6$ (**f**); and $\omega_{\text{cw}} = 0.0025$ and $\Omega = 0.2$ (**g**), $\Omega = 0.4$ (**h**), and $\Omega = 0.6$ (**i**).

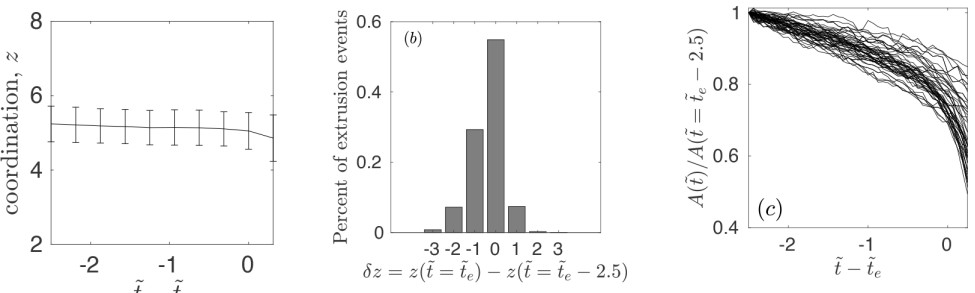

**Appendix 1—figure 3.** Coordination and area statistics of extruding cells. (**a**) The average and standard deviation of coordination number – number of interacting neighbors – for an extruding cell. The data corresponds to all simulations and the four distinct realizations considered in the article. (**b**) Percent of extrusion events with a given $z$ – characterizing the change in number of interacting neighbors at extrusion time $\tilde{t}_e$ and $\tilde{t}_e - 2.5$. The data corresponds to all simulations and the four distinct realizations considered in the article. (**c**) The temporal evolution of area for extruding cells normalized with the area at $\tilde{t}_e - 2.5$, for one of the realizations.

## Coordination number computation

To compute the coordination number, we use interaction between the cells instead of Voronoi tessellation. This is because when confluency is lost and there is a heterogeneous density of cells on the substrate, Voronoi tessellation would overestimate a cell's number of neighbors. To this end, we consider two cells, $i$ and $j$, as interacting cells if the following is satisfied:

$$\{\phi_i | \phi_i > 0.25\} \cap \{\phi_j | \phi_j > 0.25\} \neq \emptyset \qquad (7)$$

## Out-of-plane shear, $\sigma_{xz}$, for non-extruding cells

The fluctuations in out-of-plane shear, $\bar{\bar{\sigma}}^i_{xz}(\tilde{t}) = \langle \sigma_{xz}(\vec{x},\tilde{t})\rangle_{\vec{x}\in\mathcal{R}_i}/\langle \sigma_{xz}(\vec{x},\tilde{t})\rangle_{\vec{x}\in\mathcal{R}}$ for an extruding cell $i$ normalized by the maximum out-of-plane shear for all non-extruding cells in the same temporal window, as shown in **Appendix 1—figure 4**.

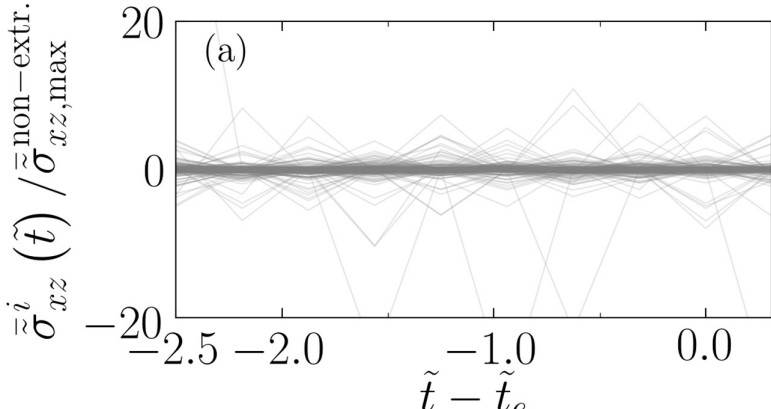

**Appendix 1—figure 4.** Oscillations of out-of-plane shear stress for extruding cells relative to non-extruding cells. (**a**) Temporal evolution of the out-of-plane shear stress for an extruding cell normalized by the maximum of the out-of-plane shear for all non-extruding cells within the same temporal window.

## Half-integer defect statistics

We have computed the lifetime for a half-integer defect by tracking that defect from its nucleation to annihilation in our simulations. The probability density for defect lifetimes are shown in **Appendix 1—figure 5**. Furthermore, we computed the defect density, which we define as the number of defects, either +1/2 or -1/2, detected at each simulation (time) frame divided by the domain of the simulation, $a = L_x \times L_y$. This is shown in **Appendix 1—figure 6**. We also computed the distances between half-integer defects for each simulation (time) frame, as shown in **Appendix 1—figure 7**. The peak in these probability densities are much larger than the peak of the minimum distance to ±1/2 defects, as shown in **Figure 2c and d**.

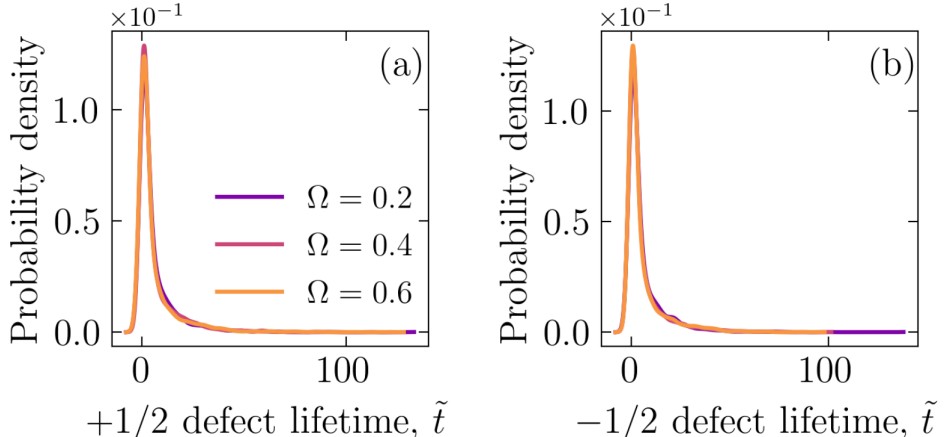

**Appendix 1—figure 5.** Defect statistics. (**a**) The probability density of +1⁄2 and (**b**) -1⁄2 defect lifetimes.

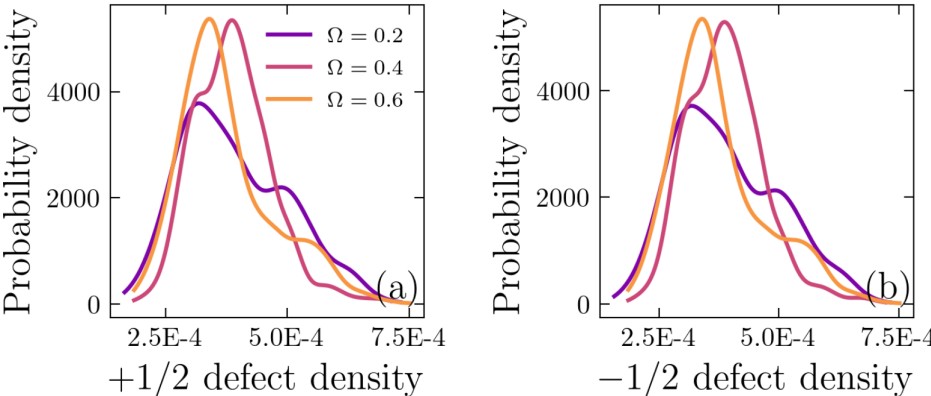

**Appendix 1—figure 6.** Probability density of (**a**) +1⁄2 and (**b**) −1⁄2 defect density distributions.

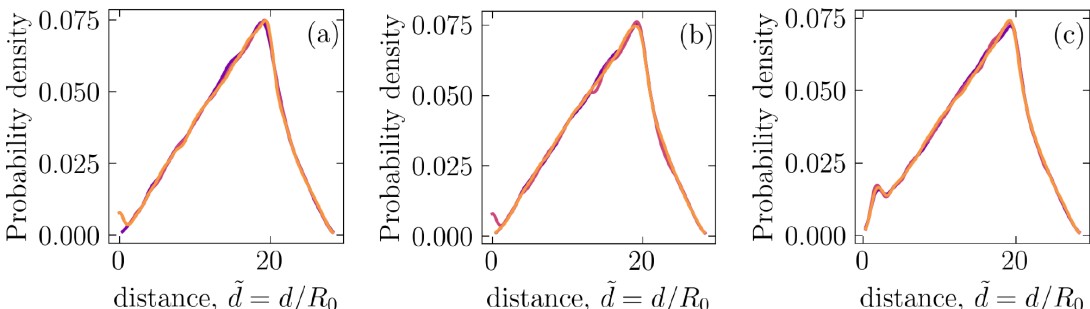

**Appendix 1—figure 7.** Probability density of pairwise distance between (**a**) +1⁄2 and +1⁄2 defects, (**b**) −1⁄2 and −1⁄2 defects, and (**c**) +1⁄2 and −1⁄2 defects.

## Phase diagram for extrusion intensity

To further explore the impact of asymmetric interaction of the cells with apical and basal sides, we have performed additional simulations varying the strength of the cell–substrate interactions. *Appendix 1—figure 8* shows how changing cell–substrate adhesion (basal), $\omega_{cw}$, affects the extrusion rate. The results show that increasing cell–substrate adhesion leads to less extrusion events, while the ratio $\Omega = \omega_{cc}/\omega_{cw}$ does not seem to play a significant role on the likelihood of an extrusion event occurring.

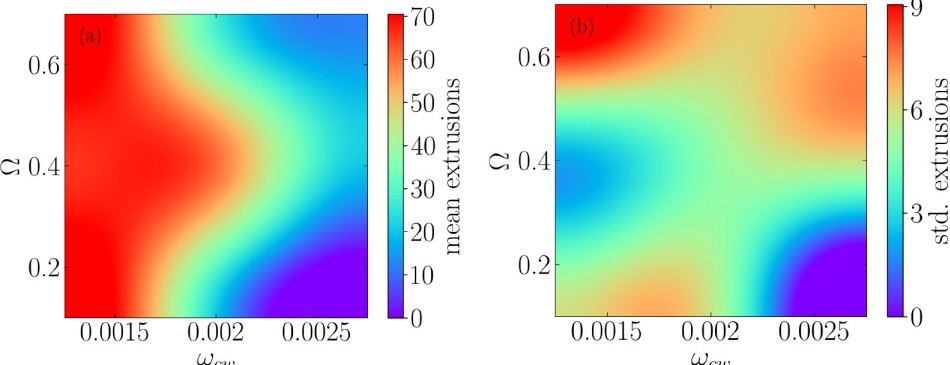

**Appendix 1—figure 8.** Extrusion phase diagram. (**a**) Mean and (**b**) standard deviation of the number of extrusions for a range of values for $\Omega = \omega_{cc}/\omega_{cw}$ and cell–substrate adhesion, $\omega_{cw}$, over the range of 29,000 times steps and for four realizations.

## Energy spectra

We calculated energy spectra for different cell-cell adhesion strengths, which suggests different power-law regimes, as shown in *Appendix 1—figure 9*. The kinetic energy spectrum, $\hat{E}_v = \frac{1}{2}\langle \hat{v}_i(k)\hat{v}_i(k)\rangle$, where $\hat{v}_i$ is the Fourier transforms of the velocity field, and $\tilde{E}_v = \hat{E}_v(k)/\hat{E}_v^{\max}(k)$. Furthermore, $\tilde{k} = k/(2\pi/R_0)$.

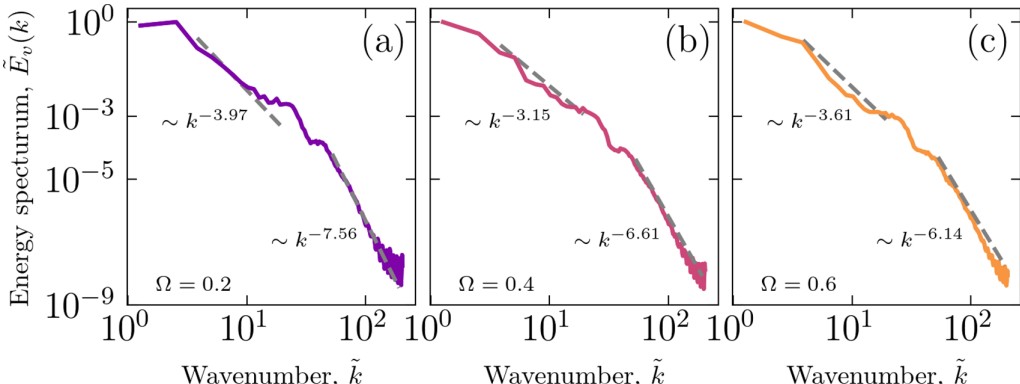

**Appendix 1—figure 9.** Energy spectra for three sample simulations with (**a**) $\Omega = 0.2$, (**b**) $\Omega = 0.4$, and (**c**) $\Omega = 0.6$.

## $p-$ atic order

We computed the $p-$ atic order parameter, associated with a liquid crystal exhibiting $p-$ fold rotational symmetry (**Nelson and Halperin, 1979**), $\psi_p^i = \frac{1}{N_n^i}\sum_j^{N_n^i}\exp(pi\theta_{ij})$, where $N_n^i$ is the number of neighbors for cell $i$, $\theta_{ij}$ is the angle between vector $\vec{r}_{ij}$, connecting cell $i$ and neighboring cell $j$, and $\vec{e}_x$. Lastly, $p = 2$ for nematic and $p = 6$ for hexatic phases. The mean of the absolute value, $|\bar{\psi}_2|$ and $|\bar{\psi}_6|$ for various $\Omega = \omega_{cc}/\omega_{cw}$ is shown in *Appendix 1—figure 10*.

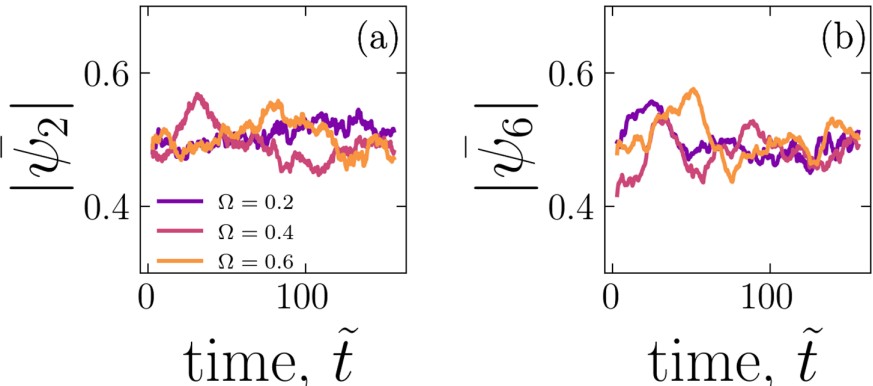

**Appendix 1—figure 10.** Temporal evolution of nematic order parameter (**a**) and hexatic order parameter (**b**) for various relative cell–cell adhesions, $\Omega$.

## Example for equilibrated monolayer configuration

In the absence of activity, cells tend to equilibrate into a hexagonal lattice. An example is shown in *Appendix 1—figure 11a* along with the temporal evolution of the mean hexatic order parameter, $|\bar{\psi}_6|$ displayed in *Appendix 1—figure 11b*, plateauing at $|\bar{\psi}_6| = 1$ indicative of perfect hexatic order.

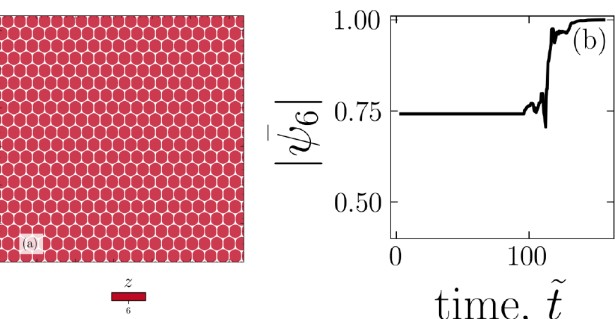

**Appendix 1—figure 11.** In the absence of active forces, cells tend to equilibrate into a hexagonal lattice. An example configuration (**a**) and evolution of the hexatic order parameter as the systems equilibrates (**b**).

