## [Editor Report]

In this work, Monfared et al. construct a valuable three-dimensional phase-field model for cell monolayers and use this to investigate the relationship between single-cell extrusion events and topological defects in cellular arrangement. The extension of existing 2D phase field models to three dimensions is an important contribution of this paper, which will be of general interest to the theoretical modelling of epithelial monolayers. Here the model is used to study the importance of cell-cell and cell-substrate interaction in extrusion from cell monolayers, which will be of practical interest to biologists and physicists working on this process. This paper presents convincing evidence that extrusion events are distinctly linked to defects in nematic and hexatic orders in the cell monolayer.

---

## [Decision Letter]

**Decision letter after peer review:**

Thank you for submitting your article "Mechanical Basis and Topological Routes to Cell Elimination" for consideration by *eLife*. Your article has been reviewed by 3 peer reviewers, and the evaluation has been overseen by a Reviewing Editor and Jonathan Cooper as the Senior Editor. The reviewers have opted to remain anonymous.

The reviewers found that the 3D phase field model you developed is a very valuable improvement over existing 2D models, but that your claims of how extrusion is linked to topological parameters was insufficiently justified.

Essential revisions:

1) Substantiate the claim that the three-dimensional phase field model is crucial for understanding cell extrusion as compared to 2D models. This should include in depth analysis of necessarily 3D parameters, such as basal and apical cell surfaces.

2) Provide a better statistics of the topological parameters and their fluctuations in normal (non extruding) cells to better assess their change during extrusion.

3) Improve the characterisation of the extrusion points, e.g. by improving the representation and discussion of the stress (Figure 4).

4) Provide a point-by-point answer to the criticism of all three reviewers and appropriately amend the paper.

*Reviewer #1 (Recommendations for the authors):*

1. The authors say "The similarities between cellular systems and liquid crystals featuring both nematic order (Saw et al., 2017; Kawaguchi et al., 2017; Duclos et al., 2018; Blanch-Mercader et al., 2018; Tan et al., 2020; Zhang et al., 2021) and hexatic order (Pasupalak et al., 2020; Maitra et al., 2020; Hoffmann et al., 2022) with the two phases potentially coexisting (Armengol-Collado et al., 2022) and interacting provide a fresh perspective for understanding cellular processes." Do the authors also find that the two phases coexist? I find this rather difficult to understand.

2. The 1/2 defects are defined using a nematic order parameter (I guess the positions of the defects are identified as the positions of zeros in this order parameter and the sign is assigned by looking at the orientation field around that). However, I think it will make the work more self-contained and easier to understand if the authors explicitly explain how they obtain a nematic order parameter field from a map of projected 2d cell shapes.

3. I am guessing that the 1/6 defects are defined purely using the co-ordination number of the cells (or do the authors construct a hexatic field and use the zeros of that to identify the hexatic disclinations?). But the probability density of average coordination number from Figure 2e seems to be peaked at 5 rather than 6. Therefore, I don't understand the rationale for considering hexatic defects (or hexatic order). Also, would it be possible to define a hexatic order parameter the same way the authors (presumably) defined a nematic order parameter (and director field) and obtain the defects explicitly as singularities in this field?

4. The authors say "Figure 4(c) shows the evolution of spatially averaged normalized isotropic stress for extruding cell…. demonstrating a clear stress build up, followed by a drop near t~−t~e=0". Maybe I am missing something, but I do not see this drop. Instead, the mean (red line) increases monotonically.5. The authors say that the out-of-plane shear stress "prior to extrusion exhibits oscillations with large magnitudes relative to the mean field." However, since they don't present any data on the usual fluctuation of the out-of-plane shear stress (i.e. for non-extruding cells), the reader has no way to judge whether this is atypical. Surprisingly, the standard deviation seems to be larger at earlier times (i.e., away from the extrusion event). Could the authors compare the standard deviation of the shear stress fluctuations of extruding and non-extruding cells?

6. If the earlier point is not clarified, it is not really clear to me what new information is obtained from a 3d model of the cell layer that couldn't be obtained from a 2d phase field model. (I do understand that their criteria for identifying extrusion is only available in the 3d model, but in a 2d model, one could choose a different criterion -- shrinkage of the (2d projected) area of a cell below a critical value or cell overlap for instance -- which would probably lead to similar results).

7. The cells in the authors' model move in the direction of the cell polarisation. It would be useful if the authors could comment how this cell polarisation is determined.

8. In the same vein, the cell polarisation doesn't directly affect the cellular structure (i.e., there is no term involving the polarisation in the free energy in Eq. 1). Is that assumption correct?

9. The authors say "Interestingly, the association of cell extrusion events with regions of high out-of-plane shear stress has parallels with the phenomenon of plithotaxis, where it was shown that cells collectively migrate along the orientation of the minimal in-plane intercellular shear stress (Tambe et al., 2011). In this context, based on the association of cell extrusion events with regions of high out-of-plane shear stress, we conjecture that high shear stress concentration hinders collective cell migration with cell extrusion providing a mechanism to re-establish the status-quo." I don't see clear evidence of high out-of-plane shear stress in 4b. Figure 4b has two extrusion sites, one of which certainly displays high out-of-plane shear stress, but the other, not so much. Could the authors quantify their claim that extrusion events are statistically associated with high out-of-plane shear stress?

10. The authors claim that the probability of extrusion at "nematic and hexatic disclinations " changes depending on cell-cell and cell-substrate adhesion. Is this a secondary consequence of the structure of the cell layer changing depending on those parameters (i.e., going from a predominantly nematic-like organisation to a more hexatic organsiation)?

11. If I understand correctly, the cells are extruded at -1/2 disclinations as well, which I find puzzling.

*Reviewer #2 (Recommendations for the authors):*

I would like to point out:

– The identification of defects, as shown in Figure 2, sensitively depends on various thresholds. The cell with 4 neighbors in Figure 2b could easily be also considered as a cell with 8 neighbors. How sensitive are the results with respect to the thresholds used?

– Why is the approach to find correlations between defects and extruding cells different for positional and orientational defects? I would find it more natural to also average over various runs and time intervals to identify such correlations for the positional defects.

– In the conclusion it is argued that negative Gaussian curvature cannot form due to the rigid substrate. This somehow indicates that the basal side is considered, for which I understand this argument. For the apical side I don't! As I assume that the experimental data in Saw et al. show the apical side, I wonder what the relation is?

*Reviewer #3 (Recommendations for the authors):*

This work explores the linkage between extrusion and topological defects in cell monolayers. To better understand this linkage, the authors used cutting-edge numerical simulations that were developed by some of the authors in Mueller et al. 2019. I have some major concerns regarding some theoretical analyses and interpretations of results (see major point 1 and minor points). I cannot recommend the publication of the present manuscript before these concerns are addressed.

1. Linkage between topological defects and extrusion events.

1.1. In line 104 to 108, it is suggested that the state in Figure 1a is an active nematic turbulent state based on the nucleation and dynamics of nematic defects. However, an active turbulent state is also characterized by specific statistical features of flows. Are the flows in simulation compatible with an active nematic turbulent state?

1.2. In line 117-124, it is suggested that extrusions in Saw et.al. 2017 correlate with the position of + and -1/2 defects. Unless I am mistaken, their observations showed spatiotemporal correlations only for +1/2 defects. Can the authors comment on this point? If this is the case, how do the authors interpret the fact that -1/2 defects also correlate with cell extrusion in their simulations? Is the mechanism for cell extrusion in -1/2 defects of mechanical origin? Adding the average isotropic stress near -1/2 defects would help to clarify this point.

1.3. In lines 124-128, the authors define a time interval around an extrusion event, but I could not understand the reason for the choice of the lower and upper values (5.625 and 0.625). How did the authors choose these values? In the Figure 4, the stress build-up occurs within a time interval of less 1 unit of time, and in Figure 8, the change in the coordination number and the area seem to occur over time intervals of 10-100 units of time. Can the authors discuss the separation of these time scales? Another relevant time scale is the mean life-time of nematic topological defects in their simulations, can discuss show the distribution of life-times of nematic defects and discuss how this is related to the other time scales?

1.4. The authors observed that extrusion preferentially occur near + and -1/2 defects. However, in Figure 2a, there are two extrusion events and tens of defects, which seems to indicate that many defects are non-functional because they do not generate an extrusion event. Is it the case? Can the authors provide an estimate of the fraction of defects that are non-functional? Along these lines, a question that remains unaddressed is whether an extrusion event can favor the nucleation of a pair of nematic defects. Can the authors comment on this?

1.5. In Figure 2a it seems that the typical separation between nematic defects is approximately 6 cell sizes. In the case that extrusion events occur randomly, I would have expected d_min to be below 3. However, in Figure 3a, d_min can reach 25 cell sizes. Can the author comment on these differences? Can the authors include the distributions of density of nematic defects in their simulation?

Besides, can the authors add numerical details in Methods on how were the extrusion events generated in the hypothesis-testing approach? For example, were cells pulled upwards with a constant force or at a constant speed?

[Editors' note: further revisions were suggested prior to acceptance, as described below.]

Thank you for resubmitting your work entitled "Mechanical Basis and Topological Routes to Cell Elimination" for further consideration by *eLife*. Your revised article has been evaluated by Jonathan Cooper (Senior Editor) and a Reviewing Editor. The referees did acknowledge that the manuscript has been significantly improved, but there remain a number of serious criticism that must be answered.

Model assumptions: Many possible 3D features that could be responsible for complex 3D cell shapes are not included in the model. This needs to be explicitly discussed in the manuscript. These include:

1) You present a 3D model and claim that it is essential to understand cell extrusion, but most of the analysis is done on 2D quantities. The results regarding the isotropic and out-of-plane shear stress are certainly interesting but do not constitute such an explanation for the extrusion process. Furthermore, a number of assumptions of the model relate to 2D features. For instance, the polarity appears to be a 2D quantity defined by an angle (probably in the x-y plane, this should be specified). This seems to exclude active out-of-plane relative movement between cells which could participate in the extrusion process in real systems. Is there any reason for the RHS of Eq. 3. to be a two-dimensional vector or is the cell velocity not strictly in the plane? This should be clarified, and the limitations of the model clearly discussed

2) Furthermore, the magnitude of the polarity vector is set to unity, which precludes fluctuations of activity among cells, a feature that is often associated with local rearrangement processes such as extrusion. This is a strong assumption that is not discussed.

2) Extrusion has often been described as the result of different mechanical properties in the basal and apical sides on the epithelia. You say in your rebuttal that this is just one element of the 3D complexity on which you choose not to focus. It is acceptable not to include this in a model in order to focus on other effects, however, this possibility should be mentioned as a relevant process in the manuscript.

3) Scutoids. It does not appear to be clearly demonstrated that scutoids are a feature of curved epithelia only, and it is quite possible that such arrangement might be relevant to the extrusion process even in flat epithelia. While a model such as the present one that does not include this possibility is certainly valuable, the text should not give the impression that they are irrelevant.

Analysis:

4) Most of the analysis is done on projected quantities, but how the projection is obtained is not clearly explained. What does the 2D representation (such as Figure 2 for instance) exactly show? Is it a cut through the epithelia at fixed z, or some kind of maximum intensity projection? This must be specified.

5) in Figure 2a, it seems that the director field (red bars) around blue +1/2 defects correspond to -1/2 defects, and vice-versa the director field around green -1/2 defects correspond to +1/2 defects. This feature is clearer for defects that are far from others. If this is indeed the case, it should be corrected.

6) In the new Figure 12, the probability density of pairwise distance between defects is presented. It is unclear whether this distance corresponds to the minimal distance between pairs of +1/2 defects or rather the distance between half-integer defects. To compare with the results from Figure 2 the former seems more appropriate. Please comment on this point and include the former distribution if necessary.

7) Some new results presented in the revised version are confusing, or insufficiently discussed. For instance the new Figure 6 in the rebuttal. If one holds cell-substrate adhesion fixed, the mean number of extrusions has a clear maximum at a particular cell-cell adhesion. However, no other quantity that the authors present seems to show a similar maximum. Maybe the origin of this is hiding in the plot of nematic defect density for different adhesions (Figure 5 of the rebuttal). If so, that plot should be improved and both that and the Figure 6 of the rebuttal should be brought to the main text, as this seems an important feature of the problem

Discussion:

8) Threshold: In your response and in the manuscript (l.192), you seem to suggest that your method does not suffer from the arbitrariness of setting a threshold to defined extrusion events, but it appears (l.99) that extrusion also involves thresholding (of the cell vertical displacement – l.99) in your case. How would this threshold influence the results?

---

## [Author Response]

Essential revisions:1) Substantiate the claim that the three-dimensional phase field model is crucial for understanding cell extrusion as compared to 2D models. This should include in depth analysis of necessarily 3D parameters, such as basal and apical cell surfaces.2) Provide a better statistics of the topological parameters and their fluctuations in normal (non extruding) cells to better assess their change during extrusion.3) Improve the characterisation of the extrusion points, e.g. by improving the representation and discussion of the stress (Figure 4).4) Provide a point-by-point answer to the criticism of all three reviewers and appropriately amend the paper.

We thank the reviewers for constructive feedback and the Reviewing Editor for summarizing the areas for improvement. What follows is our point-by-point response to the three reviewers. In summary:

We show that the localization of isotropic stresses in an extruding cell, a major point we make in our manuscript, is a three-dimensional phenomenon and cannot be captured in two-dimensional approaches (see comment 5 by reviewer 1). Furthermore, we also show the significant contrast in amplitude of out-of-plane shear oscillations in extruding cells relative to non-extruding cells (see comment 4 by reviewer 1). Also, we address comments about basal and apical properties in our studied model and provide additional data regarding the role of substrate properties on extrusion rate (see comment 3 by reviewer 2).

We have added more statistics to quantify half-integer defects, including probability densities for defect life time (see comment 3 by reviewer 3), defect density (see comment 9 by reviewer 1), and distance between defects (see comment 5 by reviewer 3) to further substantiate the claims made in our manuscript.

We have clarified some misunderstanding regarding characterization of extrusion events and their links with nematic and hexatic defects (see comment 5 by reviewer 3, comment 6 by reviewer 3, comment 17 by reviewer 3, comment 6 by reviewer 3, comment 8 by reviewer 1, comment 9 by reviewer 1).

Reviewer #1 (Recommendations for the authors):1. The authors say "The similarities between cellular systems and liquid crystals featuring both nematic order (Saw et al., 2017; Kawaguchi et al., 2017; Duclos et al., 2018; Blanch-Mercader et al., 2018; Tan et al., 2020; Zhang et al., 2021) and hexatic order (Pasupalak et al., 2020; Maitra et al., 2020; Hoffmann et al., 2022) with the two phases potentially coexisting (Armengol-Collado et al., 2022) and interacting provide a fresh perspective for understanding cellular processes." Do the authors also find that the two phases coexist? I find this rather difficult to understand.

The existence of bothtypes of defects in our simulations, namely: (i) half-integer defects in a director field based on the elongation axes of cells and (ii) the associated 5-fold and 7-fold disclinations in hexagonal packing of the cells suggests the co-existence of nematic and hexatic phases. To show this more concretely, we have now computed the *p*−atic order parameter, associated with a liquid crystal exhibiting *p*−fold rotational symmetry [1], ψpi=1Nni∑jNniexp(piθij) where Nni is the number of neighbors for cell *i*, *θ_ij_* is the angle between vector r→ij, connecting cell *i* and neighboring cell *j*, and e→x. Lastly, *p* = 2 for nematic and *p* = 6 for hexatic phases. The mean of the absolute value, ψ¯2 and ψ¯6 for various Ω = *ω_cc_/ω_cw_* is now added to the Appendix and the coexistence of nematic and hexatic order are evident from these measures.

Actions taken:

Figure 1 is added to Appendix. It details the temporal evolution of the mean nematic and hexatic order parameters.

Line 244-245: However, nematic and hexatic order parameters do not show any clear trends with 78 varying cell-cell adhesion strength Ω (see Figure 15 in Appendix).

2. I am guessing that the 1/6 defects are defined purely using the co-ordination number of the cells (or do the authors construct a hexatic field and use the zeros of that to identify the hexatic disclinations?). But the probability density of average coordination number from Figure 2e seems to be peaked at 5 rather than 6. Therefore, I don't understand the rationale for considering hexatic defects (or hexatic order). Also, would it be possible to define a hexatic order parameter the same way the authors (presumably) defined a nematic order parameter (and director field) and obtain the defects explicitly as singularities in this field?

The five- and seven-fold disclinations in the hexatic arrangement are identified based on their coordi88 nation numbers. We believe there is a misunderstanding here. Figure 2(e) is the average coordination number over a temporal window relative to extrusion, z¯, for extruded cells, not all cells. The peak in Figure 2e is therefore indication of the higher probability of extrusion near five-fold disclinations. Figure 2(e) should be contrasted to Figure 5(b), which shows the coordination number for all cells and simulation time steps, *z*, and various relative cell-cell adhesion strengths, Ω = *ω_cc_/ω_cw_*. As shown in Figure 5(b), when all cells are considered the probability of a cell having six neighbors, i.e. *z* = 6, is highest in all cases. We have added the following to the main text for further clarification:

Actions taken:

Figure 2. caption: The probability density of average coordination number ¯*z* for an extruding cell during t~=(t/t0)∈[t~e−2.5,t~e+0.3125], where t~e denotes extrusion time, *τ*_0_ = *ξR*_0_*/α* and for varying cell-substrate to cell-cell adhesion ratios Ω.

Figure 5 caption: Probability density function for the number of neighbors *z* and various Ω, for all cells and simulation time steps.

Line 167-176: To quantify the relation between extrusion events and the disclinations, the probability density of the coordination number of an extruding cell (d~min=0) averaged over the time interval, t~∈[t~e−5.625,t~e+0.625],z~, for all the realizations is shown in Figure 2(e), clearly exhibiting a sharp peak near z~=5. The coordination number is determined based on the interactions of cells (see Appendix) and this property is independent of apical or basal considerations (Kaliman et al., 2021), unlike bent epithelia. In the case of curved epithelia, a geometrical structure called scutoids can arise as a general feature with different properties associated with apical and basal sides (Galvez et al., 2018). In our setup, the asymmetric interactions of cells with apical and basal sides are captured by varying the strength of cell-substrate adhesion. In our simulations, increasing cell-substrate adhesion leads to lower extrusion events (see Figure 13 in Appendix).

3. The authors say "Figure 4(c) shows the evolution of spatially averaged normalized isotropic stress for extruding cell…. demonstrating a clear stress build up, followed by a drop near t~−t~e=0. Maybe I am missing something, but I do not see this drop. Instead, the mean (red line) increases monotonically.

As each extruding cell detaches from the substrate and loses contact with the surrounding cells, the forces due to its interactions with other cells vanish and a drop in isotropic stress is observed. To illustrate this, Author response image 1 displays one clear example with extended temporal window relative to Figure 4(c) in the main text. For further clarification we have now added explicitly the following to the main text:

**Author response image 1. sa2fig1:** The evolution of average isotropic stress for an extruding cell with extended temporal window, relative to Figure 4(c) in the main text.

Actions taken:

Line 218-218: as a cell detaches the substrate and loses contact with other cells

4. The authors say that the out-of-plane shear stress "prior to extrusion exhibits oscillations with large magnitudes relative to the mean field." However, since they don't present any data on the usual fluctuation of the out-of-plane shear stress (i.e. for non-extruding cells), the reader has no way to judge whether this is atypical. Surprisingly, the standard deviation seems to be larger at earlier times (i.e., away from the extrusion event). Could the authors compare the standard deviation of the shear stress fluctuations of extruding and non-extruding cells?

To show this more clearly we have plotted the evolution of out-of-plane shear stress, σ~¯xzi for extruding cell *i* normalized by the maximum value of the shear stress, σ~¯~xz, for *all* non-extruding cells during the same temporal window, as shown in Appendix 1—figure 9. The y-axis displays the contrast, more than an order of magnitude enhancement of the fluctuations for extruding cells, even when normalized by the maximum value within the same temporal window and for all non-extruding cells. We note that this further emphasizes the importance of the access to 3D stress fields, which is not possible with 2D models and current experimental characterization of the traction forces on 2D substrates.

Actions taken:

We have added Appendix 1—figure 9 to the Appendix. It shows the temporal evolution of the out-of-plane shear stress for an extruding cell *i*, normalized by the maximum out-of-plane shear of non-extruding cells, within the same temporal window.

Line 224-225: ...a stark departure from their non-extruding counterparts (see Figure 9 in Appendix).

5. If the earlier point is not clarified, it is not really clear to me what new information is obtained from a 3d model of the cell layer that couldn't be obtained from a 2d phase field model. (I do understand that their criteria for identifying extrusion is only available in the 3d model, but in a 2d model, one could choose a different criterion -- shrinkage of the (2d projected) area of a cell below a critical value or cell overlap for instance -- which would probably lead to similar results).

We are confident that the previous point is now clarified.

Nevertheless, we believe there is a misunderstanding here as we do not use any criterion – critical values or thresholds – to induce an extrusion event. Rather, an extrusion event generically emerges from our simulations and depends on the cell-cell and cell-substrate interactions.

We respectfully do not agree that similar results can be observed in a 2D model. As clarified in the response to the previous comment we clearly have shown the importance of having access to three dimensional fields and we have discussed thoroughly mechanisms such as buckling of hexatic defects which we would not be able to identify if we did not have access to three-dimensional fields. Furthermore, the isotropic stress build-up shown in Figure 4(c) in the main text is a three-dimensional quantity, i.e. *σ^iso^* = (1*/*3)(*σ_xx_* + *σ_yy_* + *σ_zz_*). A similar build-up is not observed in two-dimensions, i.e. *σ^iso^* = (1*/*2)(*σ_xx_* + *σ_yy_*), as shown in Author response image 2. Moreover, a decrease in cell area does not necessarily mean an extrusion event as it is well-established that projected cell area can indeed significantly fluctuate in 158 cell monolayers [2, 3]. It is then immediately clear that setting an arbitrary threshold based on the cell area reduction can lead to false extrusion events in 2D models.

**Author response image 2. sa2fig2:** The significance of the three-dimensional isotropic stress. (a) Temporal evolution of isotropic stress in two-dimensions and (b) three-dimensions for a number of extruding cells highlighting the difference between 2D and 3D isotropic stress.

6. The cells in the authors' model move in the direction of the cell polarisation. It would be useful if the authors could comment how this cell polarisation is determined.

This is explained in Materials and methods. In particular Eq. (4) outlines the dynamics of cell polarisation and immediately after this equation the polarity vector for cell *i*, p→i is defined. We have added a sentence in the main text to clarify this.

Actions taken:

Line 96-96: see Methods for polarisation dynamics.

7. In the same vein, the cell polarisation doesn't directly affect the cellular structure (i.e., there is no term involving the polarisation in the free energy in Eq. 1). Is that assumption correct?

Yes, the cell polarisation is not part of the free energy of a cell but it drives the system out of equilibrium and can influence the modes of collective self-organization of the cells and their interactions. Please see Materials and methods section for justification of the choice of polarisation mechanism based on previous (qualitative and quantitative) comparison with existing experimental results.

8. The authors say "Interestingly, the association of cell extrusion events with regions of high out-of-plane shear stress has parallels with the phenomenon of plithotaxis, where it was shown that cells collectively migrate along the orientation of the minimal in-plane intercellular shear stress (Tambe et al., 2011). In this context, based on the association of cell extrusion events with regions of high out-of-plane shear stress, we conjecture that high shear stress concentration hinders collective cell migration with cell extrusion providing a mechanism to re-establish the status-quo." I don't see clear evidence of high out-of-plane shear stress in 4b. Figure 4b has two extrusion sites, one of which certainly displays high out-of-plane shear stress, but the other, not so much. Could the authors quantify their claim that extrusion events are statistically associated with high out-of-plane shear stress?

We believe this is now fully clarified based on our response to comment 4. Figure 4(b) is just a snapshot in time. The fields (and the colormap) are normalized and the ”hotspots” are only at the two marked extrusion sites. Since there is a time gap between these two extrusion events, one extrusion site may appear to have a higher out-of-plane shear relative to the other site. However, compared to the whole domain, both sites have much larger values (please note the colorbar).

The statistics backing our claim is clearly shown in Figure 4(d). Please note the scaling of the y-axis (×10^4^) for the normalized stress. Furthermore, please see our response to comment 4 above.

9. The authors claim that the probability of extrusion at "nematic and hexatic disclinations " changes depending on cell-cell and cell-substrate adhesion. Is this a secondary consequence of the structure of the cell layer changing depending on those parameters (i.e., going from a predominantly nematic-like organisation to a more hexatic organisation)?

In Appendix 1—figure 11(b) we show that increasing relative cell-cell to cell-substrate adhesion strength Ω results in a higher probability for a cell to be at a five-fold disinclination. For completion, we have also included the half-integer defect densities for various Ω in the appendices. As shown, there are no clear trends with Ω. The density is defined as the number of ±1*/*2 defects normalized by the domain area, i.e. *a* = *L_x_* × *L_y_*. In addition, see our response to comment 1, above, where we compute mean nematic (|ψ¯2|) and hexatic (|ψ¯6|) order parameters, where we also do not observe significant variation of |ψ¯2 and |ψ¯6 with varying the cell-cell adhesion strength. Instead, we show in Appendix 1—figure 11(a) that changing cell-cell adhesion strength results in significant changes in the distribution of isotropic stress. This is the 208 potential explanation which we discuss thoroughly in the main text.

Actions taken:

We have added Appendix 1—figure 11 to the Appendix. It displays the probability density of half-integer nematic defect density for different Ω = *ω_cc_/ω_cw_*.

Line 260-261: while no clear trend is observed for the density of half-integer defects (see Figure 11 in Appendix).

10. If I understand correctly, the cells are extruded at -1/2 disclinations as well, which I find puzzling.

Yes, that is correct and consistent with Saw et al., Nature, 2017 where they observed extrusion events associated with both +1*/*2 and −1*/*2 defects, with a relatively stronger association for +1*/*2 defects. This text is from Saw et al., Nature, 2017: “We found that extrusion events were strongly correlated with the positions of a subset of +1/2 defects (and less so with -1/2 defects) (Figure 1f, Extended Data Figure 1e-h, Methods)”.

Actions taken:

Line 73-74: …found a strong correlation between extrusion events and the position of a subset of +1*/*2 defects in addition to a relatively weaker correlation with −1*/*2 defects (Saw et al., 2017).

Reviewer #2 (Recommendations for the authors):I would like to point out:– The identification of defects, as shown in Figure 2, sensitively depends on various thresholds. The cell with 4 neighbors in Figure 2b could easily be also considered as a cell with 8 neighbors. How sensitive are the results with respect to the thresholds used?

We determine the number of neighbors based on how many cells a cell is interacting with. We specifically chose this criterion as opposed to typical voronoi tessellation since during our simulations and in regions with high number of extrusion events in temporal proximity the confluency can be lost. In that case, methods based in voronoi tessellation would over-count the neighbors. To determine interactions among cells, we use the overlap between the phase-fields with a cut-off value as detailed now in the appendices.

Please note that there is only one threshold which determines the extent of the phase-field domain.

Actions taken:

We have added the following to Appendix:

To compute the coordination number, we use interaction between the cells instead of voronoi tessellation. This is because when confluency is lost and there is a heterogeneous density of cells on the substrate, voronoi tessellation would over-estimate a cell’s number of neighbors. To this end, we consider two cells, *i* and *j*, as interacting cells if the following is satisfied:

{*ϕ_i_*|*ϕ_i_ >* 0*.*25} ∩ {*ϕ_j_*|*ϕ_j_ >* 0*.*25} ≠ ∅ (1)

– Why is the approach to find correlations between defects and extruding cells different for positional and orientational defects? I would find it more natural to also average over various runs and time intervals to identify such correlations for the positional defects.

We believe this is a misunderstanding. In both cases, we look at the probability densities associated with relevant parameters. Since the half-integer defect cores associated with an extrusion event is not perfectly centered on an extruding cell, we use d~min to characterize this link. In the case of hexatic disclinations, it depends on the number of neighbors a cell is interacting with and thus an extruding cell could itself be a five-fold disclination for which d~min=0. Furthermore, similar to half-integer nematic defects (Figure 2(c)-(d)), the hextic defects in Figure 2(e) also correspond to four realization of the parametric space we probed, as also highlighted in the caption for Figure 2.

Actions taken:

Line 167-176: To quantify the relation between extrusion events and the disclinations, the probability density of the coordination number of an extruding cell (d~min=0) averaged over the time interval, t~∈[t~e−5.625,t~e+0.625],z~, for all the realizations is shown in Figure 2(e), clearly exhibiting a sharp peak near z~=5. The coordination number is determined based on the interactions of cells (see Appendix) and this property is independent of apical or basal considerations (Kaliman et al., 2021), unlike bent epithelia. In the case of curved epithelia, a geometrical structure called scutoids can arise as a general feature with different properties associated with apical and basal sides (Galvez et al., 2018). In our setup, the asymmetric interactions of cells with apical and basal sides are captured by varying the strength of cell-substrate adhesion. In our simulations, increasing cell-substrate adhesion leads to lower extrusion events (see Figure 13 in Appendix).

– In the conclusion it is argued that negative Gaussian curvature cannot form due to the rigid substrate. This somehow indicates that the basal side is considered, for which I understand this argument. For the apical side I don't! As I assume that the experimental data in Saw et al. show the apical side, I wonder what the relation is?

In our work, the cells are on a rigid substrate – that would be the basal side. The apical side is free and cells are allowed to detach from the substrate and move out-of-plane. Furthermore, we only simulate a monolayer. Thus, our arguments about hexatic defects buckling and creating positive and negative Gaussian curvatures are only relevant to the basal side. The experimental data analyses in Saw et al. 2017 is based on top view of the monolayer, while all the stress measurements are based on the traction forces on the basal side. There is no reason to believe in such a setting, a flat cell layer on a relatively rigid monolayer, there will be a significant difference in projected cell shape, whether viewed apically or basally. This is further established to be the case experimentally for a flat monolayer of epithelial cells [4]. However, this may not be the case in curved epithelia, as shown by Gomez-Galvez et al., 2018, where Scutoids could become relevant.

To further explore the impact of asymmetric interaction of the cells with apical and basal sides, we have performed additional simulations varying the strength of the cell-substrate interactions. We have added a new phase diagram showing how changing cell-substrate adhesion (basal), *ω_cw_*, affects the extrusion rate.

Actions taken:

We have added Figure 6 to Appendix. It displays the mean and standard deviation of the number of extruded cells for varying cell-substrate adhesion strength, *ω_cw_*, and relative cell-cell adhesion strength, Ω = *ω_cc_/ω_cw_*. The results correspond to four realizations of this parametric space.

Line 170-172: The coordination number is determined based on the interactions of cells (see Appendix) and this property is independent of apical or basal considerations (Kaliman et al., 2021), unlike bent epithelia.

Line 174-176: In our setup, the asymmetric interactions of cells with apical and basal sides are captured by varying the strength of cell-substrate adhesion. In our simulations, increasing cell substrate adhesion leads to lower extrusion events (see Figure 13 in Appendix).

Reviewer #3 (Recommendations for the authors):This work explores the linkage between extrusion and topological defects in cell monolayers. To better understand this linkage, the authors used cutting-edge numerical simulations that were developed by some of the authors in Mueller et al. 2019. I have some major concerns regarding some theoretical analyses and interpretations of results (see major point 1 and minor points). I cannot recommend the publication of the present manuscript before these concerns are addressed.1. Linkage between topological defects and extrusion events.1.1. In line 104 to 108, it is suggested that the state in Figure 1a is an active nematic turbulent state based on the nucleation and dynamics of nematic defects. However, an active turbulent state is also characterized by specific statistical features of flows. Are the flows in simulation compatible with an active nematic turbulent state?

Since characterizing the flow features is not the focus of the current study and could lead to deviation from the main message, we now change the terminology to “defect chaos”. Nevertheless, we acknowledge the referee’s suggestion and we have now performed calculation of energy spectra for different cell-cell adhesion strengths, which suggests different power-law regimes (see Appendix 1-figure 14). However, since there is not large enough span of wave numbers to concretely conclude a power-law scaling, we prefer to defer the analyses of the flow to future studies.

Actions taken:

Line 110-110: (see Figure 14 in Appendix for energy spectra characterization).

We have added Appendix 1-figure 14 to the Appendix. It characterizes energy spectra for various relative cell-cell adhesion strengths, Ω = *ω_cc_/ω_cw_*.

1.2. In line 117-124, it is suggested that extrusions in Saw et.al. 2017 correlate with the position of + and -1/2 defects. Unless I am mistaken, their observations showed spatiotemporal correlations only for +1/2 defects. Can the authors comment on this point? If this is the case, how do the authors interpret the fact that -1/2 defects also correlate with cell extrusion in their simulations? Is the mechanism for cell extrusion in -1/2 defects of mechanical origin? Adding the average isotropic stress near -1/2 defects would help to clarify this point.

Yes, that is correct and consistent with Saw et al., Nature, 2017 where they observed extrusion events associated with both +1*/*2 and −1*/*2 defects, with a relatively stronger association for +1*/*2 defects. This text is from Saw et al., Nature, 2017: “We found that extrusion events were strongly correlated with the positions of a subset of +1/2 defects (and less so with -1/2 defects) (Figure 1f, Extended Data Figure 1e-h, Methods)**”**. The mechanism for extrusion close to −1*/*2 defects in experiments is suggested to be related to the compression patterns that form not exactly at the defect core, but at a distance from it, as is evident from the average isotropic stress measured in experiments and in the phase field model.

**Author response image 3. sa2fig3:** Comparison of the isotropic stress patterns around −1*/*2 defects. The emergence of compressive regions away from the core, but around the arms of the defect is evident. The colormap shows isotropic stress normalized by its maximum value. The patterns corresponding to Experiments and the Continuum model of active nematics are adapted from Saw et al., Nature (2017).

Actions taken:

Line 73-74: …found a strong correlation between extrusion events and the position of a subset of +1*/*2 defects in addition to a relatively weaker correlation with −1*/*2 defects (Saw et al., 2017).

1.3. In lines 124-128, the authors define a time interval around an extrusion event, but I could not understand the reason for the choice of the lower and upper values (5.625 and 0.625). How did the authors choose these values? In the Figure 4, the stress build-up occurs within a time interval of less 1 unit of time, and in Figure 8, the change in the coordination number and the area seem to occur over time intervals of 10-100 units of time. Can the authors discuss the separation of these time scales? Another relevant time scale is the mean life-time of nematic topological defects in their simulations, can discuss show the distribution of life-times of nematic defects and discuss how this is related to the other time scales?

The time axis for Figure 4(c)-4(d) are normalized, i.e. *t*˜= *t/τ*_0_ where *τ*_0_ = *ξR*_0_*/α*, *ξ* is friction, *R*_0_ is the initial cell radius and *α* is the strength of self-propulsion forces. However, the time axis for Figure 8 is not normalized but it does correspond to exactly the same temporal window. We have now made sure that this is represented consistently to show that there are no separation of time scales. This temporal window is based on the first moment of the defect lifetime distributions, which we have now added for completion (see Appendix 1-figure 10). As shown, the first moment of defect lifetimes ⟨*t*˜^±1*/*2^⟩ ≈ 5*.*69.

Actions taken:

We have updated Figure 8 with normalized time axes, *t*˜.

Line 130-132: This temporal window is chosen based on the first moment of a defect’s lifetime distribution (see Figure 10 in the Appendix).

We have added Appendix 1-figure 10 to the Appendix. It displays the probability density of half-integer defect lifetimes.

1.4. The authors observed that extrusion preferentially occur near + and -1/2 defects. However, in Figure 2a, there are two extrusion events and tens of defects, which seems to indicate that many defects are non-functional because they do not generate an extrusion event. Is it the case? Can the authors provide an estimate of the fraction of defects that are non-functional? Along these lines, a question that remains unaddressed is whether an extrusion event can favor the nucleation of a pair of nematic defects. Can the authors comment on this?

We disagree with the view that a defect not in the vicinity of an extrusion event is “non-functional”, keeping in mind that Figure 2(a) is just a snapshot in time for one of the cases we considered. We know well that these defects are interacting, attracted and repelled by each other. As a consequence, those interactions affect the modes of collective self-organization of the cells and consequently their mechanical interactions.

By performing a laser ablation induced cell removal, it was shown experimentally in Saw et al. 2017 that extrusion does not favor the nucleation of a pair of nematic defects. We have clarified this point in the revised manuscript.

Actions taken:

Line 139-142: Furthermore, laser ablation experiments have established that an induced extrusion event does not favor the nucleation of a pair of nematic defects (Saw et al., 2017).

1.5. In Figure 2a it seems that the typical separation between nematic defects is approximately 6 cell sizes. In the case that extrusion events occur randomly, I would have expected d_min to be below 3. However, in Figure 3a, d_min can reach 25 cell sizes. Can the author comment on these differences? Can the authors include the distributions of density of nematic defects in their simulation?

Respectfully, there are a number of misunderstandings here, which we address one by one: (i) the typical distance between two nematic defects is much larger than a typical distance between an extrusion event and a half-integer nematic defect, (ii) determining what is a typical distance requires statistics and cannot be determined from one simulation snapshot as suggested here re. Figure 2(a). We now provide this data for completeness. (iii) A typical d¯min based on the probability density functions (PDF) shown in Figure 2(c) and Figure 2(d) is ≈ 1*.*5 cell size. This is based on the peak of these PDFs. (iv) Figure 3(a) corresponds to a specific hypothesis that we outlined and falsified in the text regarding the randomness of the proximity of extrusion sites and half-integer nematic defects. The case where d¯min reaches 25 cell sizes corresponds to the case where extrusions were generated using a poisson point process. In the same Figure 3(a), the results corresponding to our simulations (solid black line) peaks near 1.5 cell size. Hope these points clarify any misunderstanding of the data we present in our manuscript.

We have also added the density of defects in the appendices as requested.

Actions taken:

See our response to reviewer 1, comment 9 for the density of defects (Figure 5).

Line 136-138: …at a much smaller distance relative to a typical distance between two defects (see Figure 12 in Appendix).

We have added Figure 10 to Appendix. It characterizes the distance between half-integer nematic defects during the simulations.

Besides, can the authors add numerical details in Methods on how were the extrusion events generated in the hypothesis-testing approach? For example, were cells pulled upwards with a constant force or at a constant speed?

The goal of this hypothesis testing approach is to show that the correlation between +1*/*2 defects and the extrusion events is not random. To provide a null hypothesis we generated extrusion event locations. Specifically, we use a poisson point process to randomly generate (x,y) coordinate of an extrusion location. Furthermore, we use a uniform distribution to place this random extrusion event at a random time step. Then, we carry out our analysis based on the nematic defects in that simulation time frame. Thus, there are no physical criterion for inducing an extrusion, e.g. constant force or speed. We note that the same approach is used in the experiments for hypothesis testing to show that the correlations observed between defects and extrusions are significant compared to defects correlations with random point within the tissue.

Actions taken:

Line 149-149: …randomly generate positions for extrusion events a…

[Editors' note: further revisions were suggested prior to acceptance, as described below.]

The referees did acknowledge that the manuscript has been significantly improved, but there remain a number of serious criticism that must be answered first.Model assumptions: Many possible 3D features that could be responsible for complex 3D cell shapes are not included in the model. This needs to be explicitly discussed in the manuscript. These include:1) You present a 3D model and claim that it is essential to understand cell extrusion, but most of the analysis is done on 2D quantities. The results regarding the isotropic and out-of-plane shear stress are certainly interesting but do not constitute such an explanation for the extrusion process. Furthermore, a number of assumptions of the model relate to 2D features. For instance, the polarity appears to be a 2D quantity defined by an angle (probably in the x-y plane, this should be specified). This seems to exclude active out-of-plane relative movement between cells which could participate in the extrusion process in real systems. Is there any reason for the RHS of Eq. 3. to be a two-dimensional vector or is the cell velocity not strictly in the plane? This should be clarified, and the limitations of the model clearly discussed.

Polarity in our current model is front-rear polarity and it only acts in plane. However, the in-plane active self-propulsion force associated with cell polarity can induce forces and velocities with non-zero out-of-plane components, complementing forces due to other interactions e.g. to cell-cell adhesion, as a cell described by ∅i(χ→) deforms in three-dimensions. The velocity field is a three-dimensional vector field, as shown in Figure 1(b) and thus includes out-of-plane components. Furthermore, we have established before (see our response to Reviewer 1, comment 5 of our previously submitted rebuttal), we define pressure as a three-dimensional quantity p=(13)(σxx+σyy+σzz) rather than a two-dimensional one p=(12)(σxx+σyy). These points have been clarified in the text.

Actions taken:

Line 353-356: It is worth noting that the self-propulsion forces, F→isp, associated with cell polarity, p→i, 50 acts in-plane but can induce out-of-plane components in force and velocity fields as a cell described 51 by ∅i(χ→) deforms in three-dimensions (see Equation (3)).

Line 351: p→i=(cos⁡θi,sin⁡θi.0)

Line 96: front-rear

Line 347: front-rear

2) Furthermore, the magnitude of the polarity vector is set to unity, which precludes fluctuations of activity among cells, a feature that is often associated with local rearrangement processes such as extrusion. This is a strong assumption that is not discussed.

As discussed in the text, the activity in our model is due to self-propulsion forces (F→isp) associated with cell polarity p→i. The dynamics of polarisation is given in Equation (4) which includes a Gaussian white noise (*η*) and a rotational diffusivity constant (*D_r_*) leading to fluctuations in the direction of polarity and thus self-propulsion forces via F→isp=αp→i, where *α* represents the strength of cell motility.

Actions taken:

Line 353-356: It is worth noting that the self-propulsion forces, F→isp, associated with cell polarity, p→i, acts in-plane but can induce out-of-plane components in force and velocity fields as a cell described by ∅i(χ→) deforms in three-dimensions (see Equation (3)).

3) Extrusion has often been described as the result of different mechanical properties in the basal and apical sides on the epithelia. You say in your rebuttal that this is just one element of the 3D complexity on which you choose not to focus. It is acceptable not to include this in a model in order to focus on other effects, however, this possibility should be mentioned as a relevant process in the manuscript.

We now have explicitly mentioned this possibility as a topic of interest for future studies.

Actions taken:

Line 277-278: In the future, it can be illuminating to study the effect of heterogeneity in the apical basal mechanical response due to different mechanical properties and/or the nature of activity.

4) Scutoids. It does not appear to be clearly demonstrated that scutoids are a feature of curved epithelia only, and it is quite possible that such arrangement might be relevant to the extrusion process even in flat epithelia. While a model such as the present one that does not include this possibility is certainly valuable, the text should not give the impression that they are irrelevant.

We have updated our description of scutoids in the text, first suggested by one of the reviewers to be included in our manuscript, and have simplified it as to avoid any impression about its relevance or irrelevance in flat geometry.

Actions taken:

Line 173: geometrical structures called scutoids that have been identified in curved epithelial tubes (Gómez-Gálvez et al., 2018).

Analysis:5) Most of the analysis is done on projected quantities, but how the projection is obtained is not clearly explained. What does the 2D representation (such as Figure 2 for instance) exactly show? Is it a cut through the epithelia at fixed z, or some kind of maximum intensity projection? This must be specified.

The projected fields (Figures 2(a), 4(a)-(b), 5(c)) correspond to the basal side, i.e. the layer immediately on top of the substrate. This choice is based on (a) most of the experimental analyses are typically done on the basal side (see e.g. Balasubramaniam et al., 2021 [1]) and (b) we are interested in the interplay of cell-substrate and cell-cell adhesions.

Actions taken:

Figure 4 caption: projected into *xy*− plane (*z* = 0, i.e. the basal side)

Figure 5 caption: projected into *xy*− plane with *z* = 0 i.e. the basal side

Line 107: (*z* = 0, i.e. the basal side)

6) in Figure 2a, it seems that the director field (red bars) around blue +1/2 defects correspond to -1/2 defects, and vice-versa the director field around green -1/2 defects correspond to +1/2 defects. This feature is clearer for defects that are far from others. If this is indeed the case, it should be corrected.

We thank the reviewer for noticing this. We have updated the figure with correct symbols/legend.

Actions taken:

See updated Figure 2(a).

7) In the new Figure 12, the probability density of pairwise distance between defects is presented. It is unclear whether this distance corresponds to the minimal distance between pairs of +1/2 defects or rather the distance between half-integer defects. To compare with the results from Figure 2 the former seems more appropriate. Please comment on this point and include the former distribution if necessary.

This is explained in the Appendix (lines 619-621). Figure 12 shows the probability density of distance between half-integer defects, specifically the pairwise distance between +1*/*2 and +1*/*2 defects (Figure 12(a)) and −1*/*2 and −1*/*2 defects (Figure 12(b)). We have also added an additional plot for the pairwise distance between +1*/*2 and −1*/*2 defects displaying a similar behavior (Figure 12(c)).

Actions taken:

Figure 12(c) has been added.

Figure 12 in the Appendix caption: Probability density of pairwise distance between (a) +1*/*2 and +1*/*2 defects, (b) −1*/*2 and −1*/*2 defects and (c) +1*/*2 and −1*/*2 defects.

8) Some new results presented in the revised version are confusing, or insufficiently discussed. For instance the new Figure 6 in the rebuttal. If one holds cell-substrate adhesion fixed, the mean number of extrusions has a clear maximum at a particular cell-cell adhesion. However, no other quantity that the authors present seems to show a similar maximum. Maybe the origin of this is hiding in the plot of nematic defect density for different adhesions (Figure 5 of the rebuttal). If so, that plot should be improved and both that and the Figure 6 of the rebuttal should be brought to the main text, as this seems an important feature of the problem

We thank the reviewer for their thoughts on the newly presented figures. The mentioned maximum is only present for one value of cell-substrate adhesion (*ω_cw_* = 0*.*002). Furthermore, this maximum becomes less pronounced considering the standard deviation in extrusions as shown in Figure 6(b). Lastly, the probability densities shown in Figure 5 of the rebuttal correspond to data sorted by the relative cell-cell adhesion Ω and not cell-substrate adhesion *ω_cw_*.

Discussion:9) Threshold: In your response and in the manuscript (l.192), you seem to suggest that your method does not suffer from the arbitrariness of setting a threshold to defined extrusion events, but it appears (l.99) that extrusion also involves thresholding (of the cell vertical displacement – l.99) in your case. How would this threshold influence the results?

There is a misunderstanding here as we have been very careful with this in the text. To reiterate, our method does not suffer from the arbitrariness of setting a threshold to defined extrusion events. The extrusion events emerge from the collective interactions of the cells and involve a cell detaching from the substrate. l.192 refers to what we use to *detect* (≠ define) an extrusion event as we *post-process* the simulation results. This is a sufficient criteria as we have verified it through visualization of the simulations and a more computationally expensive method of the intersection of cell domains with the substrate.

References

[1] Lakshmi Balasubramaniam, Amin Doostmohammadi, Thuan Beng Saw, Gautham Hari Narayana Sankara Narayana, Romain Mueller, Tien Dang, Minnah Thomas, Shafali Gupta, Surabhi Sonam, Α S. Yap, Yusuke Toyama, Ren´e-Marc M`ege, Julia M. Yeomans, and Benoˆıt Ladoux. Investigating the nature of active forces in tissues reveals how contractile cells can form extensile monolayers. *Nature Materials*, February 2021.